# A distributed simple dynamical systems approach (dS2 v1.0) for computationally efficient hydrological modelling at high spatio-temporal resolution

Joost Buitink[1], Lieke A. Melsen[1], James W. Kirchner[2, 3, 4], and Adriaan J. Teuling[1]

[1]Hydrology and Quantitative Water Management Group, Wageningen University, Wageningen, The Netherlands
[2]Department of Environmental Systems Science, ETH Zurich, Zurich, 8092, Switzerland
[3]Swiss Federal Research Institute WSL, Birmensdorf, 8903, Switzerland
[4]Department of Earth and Planetary Science, University of California, Berkeley, California, 94720, USA

**Correspondence:** Joost Buitink (joost.buitink@wur.nl)

**Abstract.** In this paper, we introduce a new numerically robust distributed rainfall runoff model for computationally efficient simulation at high spatio-temporal resolution: the distributed simple dynamical systems (dS2) model. The model is based on the simple dynamical systems approach as proposed by Kirchner (2009), and the distributed implementation allows for spatial heterogeneity in the parameters and/or model forcing fields at high spatio-temporal resolution (for instance as derived from precipitation radar data). The concept is extended with snow and routing modules, where the latter transports water from each pixel to the catchment outlet. The sensitivity function, which links changes in storage to changes in discharge, is implemented by a new 3-parameter equation that is able to represent the widely observed downward curvature in log–log space. The simplicity of the underlying concept allows the model to calculate discharge in a computationally efficient manner, even at high temporal and spatial resolution, while maintaining proven model performance. The model code is written in Python in order to be easily readable and adjustable while maintaining computational efficiency. Since this model has short run times, it allows for extended sensitivity and uncertainty studies with relatively low computational costs. A test application shows a good and constant model performance across scales ranging from 3 to over 1700 $km^2$.

## 1 Introduction

Hydrological models are essential tools for applications ranging from sensitivity analysis to impact assessment and forecasting. Generally, the aim of rainfall–runoff models is to simulate streamflow given precipitation input. Depending on factors such as the research aim and the climatological/geological setting, different model structures or process representations might be preferred. This, in combination with the inherent complexity and heterogeneity of (sub)surface hydrological processes, has led to the development of numerous different hydrological models over the past decades, each with its own focus. Examples of such rainfall–runoff models and modelling tools include, amongst many others: SWAT (Arnold et al., 1998), HBV (Lindström et al., 1997), TOPMODEL (Beven and Kirkby, 1979), VIC (Liang et al., 1994, 1996), SPHY (Terink et al., 2015), FUSE (Clark et al., 2008), SUMMA (Clark et al., 2015), PCR-GLOBWB (Sutanudjaja et al., 2018) and WALRUS (Brauer et al., 2014). Although there is an ongoing debate about whether the hydrological modelling community should move towards a

community model (Weiler and Beven, 2015), different models representing a wide range of complexity and different process representations might be necessary to adequately characterise uncertainty.

Hydrological models are often classified from more conceptual to more process-based models. Conceptual models have fewer processes explicitly parametrized, and as a result their limited number of parameters makes them easier to calibrate. In conceptual models, catchments are often represented by a series of buckets or storages, which mimic processes with different response times. Their typical scale of application is that of small to medium (mesoscale) catchments, generally in a lumped fashion. Process-based models, on the other hand, contain many more parameters. These models are often applied in a distributed fashion, where many of the parameter values are based on maps of vegetation and soil properties. However, often conceptual parameters remain that require calibration or tweaking. Even though process-based models should give a better representation of the physical reality, many models can easily be beaten in performance by a simple neural network (Abramowitz, 2005), or model results can be reproduced without much loss of accuracy by models with a much smaller number of parameters (Koster and Suarez, 2001; Best et al., 2015; Liu et al., 2018). So if the research aim does not require the use of complex or specific process representation, simple models will likely outperform more complex models in many applications.

Whereas conceptual models often perform satisfactorily at the daily resolution at the lumped basin scale, they lack the ability to explicitly simulate spatially distributed processes that might be needed to accurately predict streamflow response at larger scales. In a study over a large number of basins in France, Lobligeois et al. (2014) found that (hourly) model performance markedly increased from the (lumped) basin-scale to a resolution of around 10 km$^2$ when using aggregated radar precipitation as model input. Other studies (e.g. Ruiz-Villanueva et al., 2012) have also highlighted the importance of spatial variability of rainfall, in particular the movement of storms with respect to the channel network, for flash flood simulation. Another example of a spatially distributed process that affects streamflow is the melting of snow that can depend both on elevation (via temperature) and aspect (via radiation). Comola et al. (2015), for example, showed that aspect needs to be considered for accurate simulation of snowmelt and runoff dynamics in mesoscale catchments. The spatial organization of the stream network within a basin also affects the response to rainfall, as has been shown by studies based on the catchment width function or the Geomorphological Instantaneous Unit Hydrograph (Kirkby, 1976; Rodríguez-Iturbe and Valdes, 1979). Thus, a spatial resolution in the order of 1–5 km to capture effects of rainfall variability and stream network organization, combined with a temporal resolution of 1 h or less to capture individual (convective) rain storms, is necessary for realistic rainfall-runoff modelling at larger scales.

The need to improve spatially explicit information in hydrological models aligns with the increasing availability of high-resolution continental-scale forcing datasets. These include for instance merged radar data (e.g Huuskonen et al., 2013; Winterrath et al., 2018), interpolated station data (e.g. van Osnabrugge et al., 2017; Cornes et al., 2018; van Osnabrugge et al., 2019), or atmospheric reanalysis products (e.g Albergel et al., 2018). These datasets have a spatial resolution in the order of one kilometre and a temporal resolution in the order of one hour. There is a growing need for easy–to–apply and easy–to–adjust models that can exploit the potential of spatially distributed input data at high spatial and temporal resolution.

Conceptual models have been applied in a (semi-)distributed manner to account for spatially distributed input data: a lumped model is applied at each individual grid cell, and water is most often transferred to the outlet using a post-processing function

that accounts for catchment routing. This method of running the model subsequently for each individual grid cell is, however, not necessarily the most computationally efficient way to deal with spatially distributed data, but often the result of historical developments. Whereas increased computational power has driven the application of distributed models at increasingly fine (spatial) resolution (Melsen et al., 2016b), it has also led to new challenges: many aspects of distributed modelling, such as

uncertainty estimation and (spatial) parameter estimation, require a large number of runs which further increases the computational demand. This demand is in the vast majority of cases too high to be tackled by individual computers, and thus computational infrastructure such as high performance clusters are used. However, there are costs related to this procedure, while the "free" computational power in the individual computer could be used more optimally. A conceptual model that can deal with high resolution gridded data to tackle these kind of issues in an efficient manner is currently lacking.

An example of a conceptual model that, in spite of its extreme simplicity, has shown good performance for discharge simulation at the scale of smaller catchments and at fine temporal (hourly) resolution is the simple dynamical systems (SDS) approach introduced by Kirchner (2009). This concept is based on the assumption that discharge is solely dependent on the total amount of stored water in a catchment. It translates changes in storage to changes in discharge using a discharge sensitivity function, without describing internal catchment processes. This sensitivity function is typically parametrized by a 2-parameter

powerlaw, however several studies have suggested a more complex downward-curving rather than linear behaviour in double-logarithmic space (Kirchner, 2009; Teuling et al., 2010; Adamovic et al., 2015). Moreover, the system parameters can be inferred from streamflow recession analysis and potentially from hillslope characteristics (Troch et al., 2003), potentially removing the need for model calibration (Melsen et al., 2014). The model's simplicity has the important advantages that discharge can be simulated based on a single equation — which can easily be vectorized for distributed implementation —

and that the model only needs to store a limited number of variables, reducing the model output fields — since storage and discharge are directly linked via the discharge sensitivity function. Especially this latter part is fundamentally different from the many "bucket-based" conceptual models, which use storage in conceptual reservoirs to determine the outflow. In the original test in the humid Plynlimon catchments (area 8.70 and 10.55 km$^2$), Kirchner (2009) found Nash–Sutcliffe Efficiencies (NSE's) exceeding 0.95 during the model validation when calibrating model parameters, and efficiencies exceeding 0.90 when

parameters were obtained from recession analysis. Others have also found the concept to work well in less humid catchments. Teuling et al. (2010) found the method to generally work well in the small (3.3 km$^2$) Swiss Rietholzbach catchment. Although the method can be expected to work best in hilly catchments, Brauer et al. (2013) found the model to produce reasonable efficiencies after calibration in the small (6.5 km$^2$) Dutch Hupsel Brook catchment. Adamovic et al. (2015) reported NSE values exceeding 0.6 for most years in several Ardechian catchments in the order of 10–100 km$^2$. Given the simplicity and

good performance of the simple dynamical systems approach at the spatial (order 10 km$^2$) and temporal (1 h) resolution required for optimal simulation of runoff in larger (mesoscale) catchments, combining it with simple representations of routing and snowmelt should result in a model that satisfies the required criteria outlined in the previous paragraphs. In a first approach to apply the simple dynamical systems approach spatially, Adamovic et al. (2016) developed a semi-distributed implementation of the SDS approach (SIMPLEFLOOD), only using sub-basins to distribute the catchment and not a grid. This model, however,

was not optimized for computational efficiency and/or numerical stability over a wide range of model parameters required for optimization studies, and the code was not made available as open source.

Here, we present a flexible and computationally efficient distributed implementation and extension of the simple dynamical systems approach that can be used to investigate the spatially distributed hydrological response using data at high spatial and temporal resolution: the distributed simple dynamical systems (dS2) model. Our aim was to create a model that is capable of computationally efficient simulations of discharge in mesoscale basins at high spatio–temporal resolutions, but also to develop a model code that is easy to use, read and modify. Therefore, the model is written in Python, although there are more computationally efficient languages available. We believe, however, that this is the optimum between model speed and code adjustability required in most hydrological model studies. In this model, the catchment is divided into smaller sections using a regular grid, and discharge is simulated according to the SDS approach for each pixel. This distributed implementation allows the concept to be applied to bigger catchments, and also allows for spatial heterogeneity, both in the forcing and in the parameters. Since the original concept consists essentially of only one differential equation, it can be applied in a computationally efficient fashion, vectorizing all cells in the catchment. Snow and routing modules are added to the model to allow for application in snow-dominated regions, and to transport the water from each pixel to the catchment outlet via the drainage network. This efficient distributed implementation lowers the computational burden for high spatial and/or temporal resolution studies, and opens doors for extensive uncertainty studies. We will first introduce the model concept and describe the technical application. After that, we discuss the parameter sensitivity and finally, we show an application of the model.

## 2 Model concept

### 2.1 Simple dynamical systems approach

The simple dynamical systems approach proposed by Kirchner (2009) combines the conservation–of–mass equation (assuming a constant density) with the assumption that discharge is solely dependent on the total storage (excluding snow and ice) in the area of interest:

$$\frac{\mathrm{d}S}{\mathrm{d}t} = P - E - Q, \tag{1}$$

$$Q = f(S), \tag{2}$$

where $S$ represents the total storage, $P$ the precipitation, $E$ the actual evaporation and $Q$ the discharge. Differentiating Eq. (2) and combining with Eq. (1) results in the following equation:

$$\frac{\mathrm{d}Q}{\mathrm{d}t} = \frac{\mathrm{d}Q}{\mathrm{d}S}\frac{\mathrm{d}S}{\mathrm{d}t} = \frac{\mathrm{d}Q}{\mathrm{d}S}(P - E - Q) = g(Q)(P - E - Q), \tag{3}$$

where we define $\frac{\mathrm{d}Q}{\mathrm{d}S}$ as the sensitivity function $g(Q)$, describing the sensitivity of discharge to changes in storage. Note that the evaporation represents actual evaporation, as the concept does not simulate evaporation reduction as result of e.g. soil moisture stress during dry periods. Ideally, a soil moisture model would provide the correct actual evaporation values, or when

applying the concept to humid areas, the actual evaporation is equal to a factor times the potential evaporation. To support the latter, we introduce the evaporation correction parameter $\epsilon$, which can correct the provided (potential) evaporation input data ($E = \epsilon \cdot E_{\text{input}}$). Additionally, to prevent numerical issues, a simple evaporation reduction switch is added, which is described in Section 3.2.

Originally, the SDS approach was introduced as a lumped approach to simulate discharge in small catchments of approximately 10 km$^2$ (Kirchner, 2009). Several studies have subsequently applied this concept to other catchments in Europe (Teuling et al., 2010; Krier et al., 2012; Brauer et al., 2013; Melsen et al., 2014; Adamovic et al., 2015). Some of the catchments in these studies had a size similar to the original scale from Kirchner (2009), yet it was also applied to catchments up to 1000 km$^2$ in size. One could argue whether the concept is still valid at a scale so different from the scale for which it was initially developed

(Beven, 1989, 2001; Sivapalan, 2006; McDonnell et al., 2007), and whether a single sensitivity function is sufficient to capture the spatial complexity of substantially larger basins. In regions with high spatial heterogeneity, grid-based models are likely to yield more realistic results than lumped models (Lobligeois et al., 2014).

To respect the original scale of development and to capture spatial variability, we have developed a distributed implementation of the simple dynamical systems approach. Our distributed implementation builds on the original concept as proposed

by Kirchner (2009), and extends this concept with simple snow and routing modules. For the distributed implementation, we assume that the SDS approach is valid for each pixel of a rectangular grid. By defining pixels with a size corresponding to the original scale (in the order of 1 km$^2$), the scale of application remains similar to the original scale, and both the forcing and the model parameters can be defined for each individual pixel (see Figure 1). In this distributed implementation, we allow precipitation to fall as snow; see Section 2.3. We added a routing module to transport water from each pixel to the river outlet,

by adding a time delay to each pixel based on the distance to the outlet and a travel speed parameter; see Section 2.4. The model can be run with different choices for $\Delta t$, yet in order to respect the spatio–temporal resolution the default time step is one hour. This model has been built with a focus on computational efficiency, meaning that all grid cells in the catchment are stored in a single vector, allowing for vectorized computations (see Fig. 1b). This results in a matrix with the rows and columns indicating time and space, respectively. As a result, the routing conceptualization is a modification of this matrix, where each

column is shifted to induce a temporal delay.

## 2.2  Discharge sensitivity

As previously mentioned, the sensitivity function is required to translate changes in storage to changes in discharge. Kirchner (2009) originally presented a simple power–law version of this sensitivity function for purposes of illustration:

$$g(Q) = \frac{\mathrm{d}Q}{\mathrm{d}S} = aQ^b, \tag{4}$$

where $a$ and $b$ define the slope and intercept of the sensitivity function in log–log space. This power–law relation has been widely used in experimental and theoretical studies (e.g. Troch et al., 1993; Brutsaert and Lopez, 1998; Tague and Grant, 2004; Rupp and Selker, 2006; Lyon and Troch, 2007; Rupp and Woods, 2008). This relatively simple sensitivity function allows the

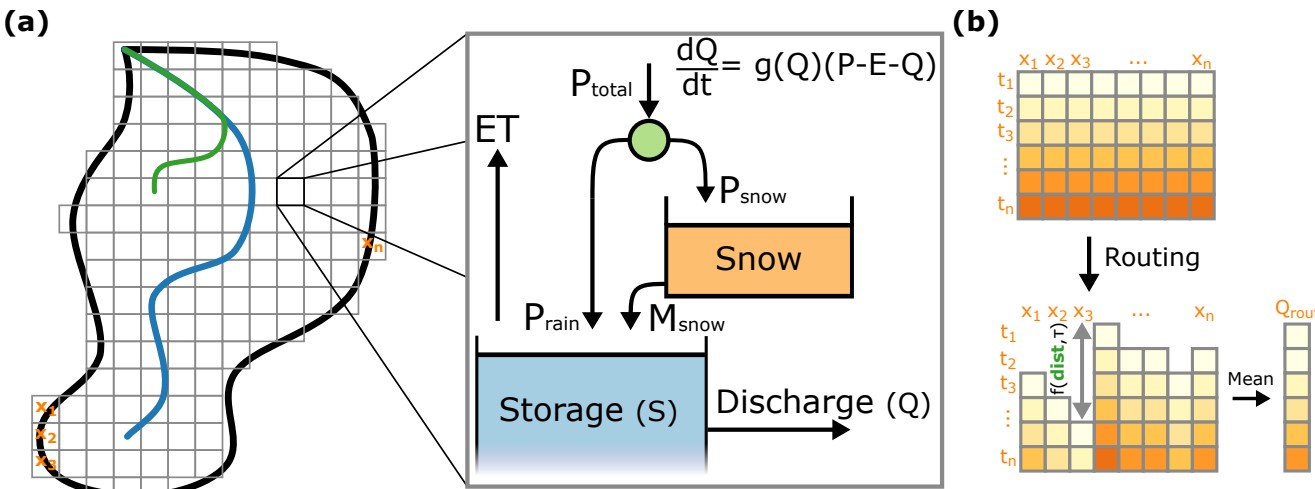

**Figure 1.** Efficient distributed implementation of the simple dynamical systems approach, which is solved for each pixel. The left hand side of panel a represents a catchment, with a regular grid covering the catchment area. The green line indicates the flow path on which the routing lag is based (green "dist" in panel b). Panel b shows how the catchment is translated to a matrix to allow for computationally efficient calculations, and how the matrix is modified in the routing algorithm, by shifting each column based on the distance to the outlet and the routing parameter $\tau$.

translation of discharge into storage, using the following equation:

$$\int dS = \int \frac{dQ}{g(Q)}, \tag{5}$$

$$S(Q) = \begin{cases} \dfrac{1}{a}\dfrac{1}{1-b}Q^{1-b} + S_0, & b \neq 1 \\ \dfrac{1}{a}\ln(Q) + S_0, & b = 1, \end{cases} \tag{6}$$

where $S_0$ is the integration constant, meaning that only relative storage changes can be obtained using this method. The value
5   of $b$ affects the meaning of $S_0$, as described by Kirchner (2009). However, most catchments show recession behaviour that differs from a power–law relation between $dQ/dS$ and $Q$ (Kirchner, 2009; Teuling et al., 2010; Krier et al., 2012; Adamovic et al., 2015). Therefore, a more complex formulation of the sensitivity function was also proposed by Kirchner (2009):

$$\ln(g(Q)) = c_1 + c_2 \ln(Q) + c_3 (\ln(Q))^2, \tag{7}$$

where $c_1$, $c_2$ and $c_3$ are the three parameters of this quadratic equation. This quadratic equation allows for a concave relation
10   between the recession rate and the discharge in log–log space. This equation has one downside, however: since it is shaped like a parabola in log–log space, there is always an optimum in the discharge sensitivity. This implies that with an increasing $Q$, the system becomes less sensitive at some point. This behaviour is unrealistic and unwanted when performing automatic calibration or random parameter sampling runs. Therefore, we have added an additional term to the original power–law equation, which

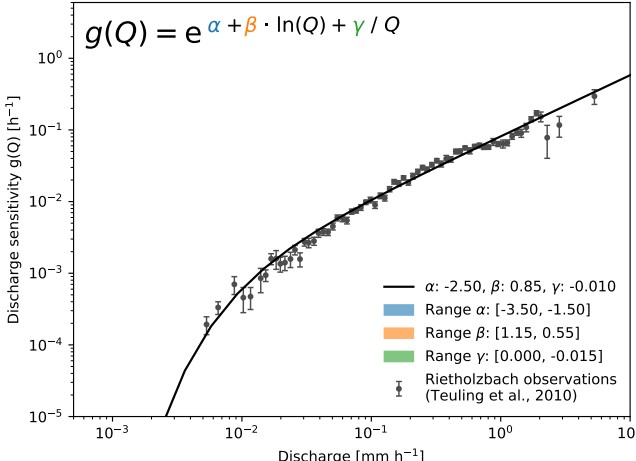

**Figure 2.** Parametrization of the discharge sensitivity, including the effects of the three parameters. $\alpha$ affects the intercept, $\beta$ affects the slope and $\gamma$ affects the curvature at low discharge values. Parameter values for each parameter are given in the legend as their min–max range.

accounts for the concave shape of the sensitivity function:

$$g(Q) = aQ^b \cdot \mathrm{e}^{\gamma/Q}, \tag{8}$$

where $a$, $b$ and $\gamma$ are the three parameters describing the shape of the discharge sensitivity. However, we have rewritten this equation to include all parameters in the exponent term, as this improves computational efficiency:

$$g(Q) = \mathrm{e}^{\alpha+\beta\cdot\ln(Q)+\gamma/Q}, \tag{9}$$

with $\alpha = \ln(a)$ and $\beta = b$ from the Eq. (8). In Fig. 2, the effect of each parameter on the shape of the sensitivity function is presented. Discharge observations from Teuling et al. (2010) are also included to indicate the importance of the $\gamma$ parameter. This equation has the benefit that it can be rewritten to the original power law equation, if $\gamma = 0$:

$$g(Q) = \mathrm{e}^{\alpha+\beta\cdot\ln(Q)} = \mathrm{e}^{\alpha} \cdot Q^{\beta}, \tag{10}$$

where $a = \mathrm{e}^{\alpha}$ and $b = \beta$ from Eq. (4). This concept allows us to use parameters from previous studies, and to transform the discharge time series to a storage time series according to Eq. (5). A disadvantage of this new sensitivity function is that, due to the addition of the $\gamma/Q$ term, there is no longer an analytical solution, but the equation can still be integrated numerically.

### 2.3 Snow processes

In our distributed implementation, we allow precipitation to fall as snow. Snow is treated as a separate storage (see Fig.
1a), where snowmelt is added to the simple dynamical systems approach in the form of liquid precipitation, following the methodology of Teuling et al. (2010). We assume snowmelt to be dependent on both temperature and radiation, following

the restricted degree–day radiation balance approach (Kustas et al., 1994). The snow storage is conceptualized based on the following equations:

$$\frac{\mathrm{d}S_{\text{snow}}}{\mathrm{d}t} = P_{\text{snow}} - M_{\text{snow}}, \tag{11}$$

$$P_{\text{snow}} = \begin{cases} P_{\text{total}} & \text{if } T <= T_0 \\ 0 & \text{if } T > T_0, \end{cases} \tag{12}$$

$$M_{\text{snow}} = \begin{cases} ddf \cdot (T - T_0) + rdf \cdot R_{\text{g}} & \text{if } M_{\text{snow}} \cdot \Delta t <= S_{\text{snow}} \\ \dfrac{S_{\text{snow}}}{\Delta t} & \text{if } M_{\text{snow}} \cdot \Delta t > S_{\text{snow}}, \end{cases} \tag{13}$$

where $S_{\text{snow}}$ is the total snow storage in mm, $P_{\text{snow}}$ the precipitation falling as snow in mm h$^{-1}$, $M_{\text{snow}}$ is the snowmelt in mm h$^{-1}$, $T$ is the air temperature in °C, $T_0$ is the critical temperature for snowmelt in °C, $ddf$ is the degree-day factor in mm h$^{-1}$ °C$^{-1}$, $rdf$ is the conversion factor for energy flux density to snowmelt depth in mm h$^{-1}$ (W m$^{-2}$)$^{-1}$, $R_{\text{g}}$ is the global radiation in W m$^{-2}$, and $\Delta t$ is the time step in hours. If no radiation observations are available, the snowmelt equation is modified to the normal degree-day method.

In Fig. 3, the effect of this snow conceptualization is presented using synthetic forcing data. In this figure, one can see that radiation can cause snow to melt even when temperatures are still below the critical temperature ($0$°C in this example). If the temperature exceeds this threshold, radiation amplifies the melting of snow, resulting in an earlier depletion of the snow storage. Finally, if snow processes are not relevant in the region of interest, the snow conceptualization can be turned off to further reduce the computational demand (see the dashed line in Fig. 3 for the resulting discharge simulation).

## 2.4 Flow routing

A simple efficient routing conceptualization was added to the distributed model to transport water from each grid cell to the outlet of the catchment. In most studies applying the simple dynamical systems approach, a constant delay factor was added to the generated runoff time series, in order to account for the delay caused by the river network. Since our distributed implementation has pixels at distinctly different locations, we need to account for attenuation caused by the river network. The routing concept presented here is based on the already existing width function concept (Kirkby, 1976), where it is assumed that the stream network induces a temporal delay proportional to the distance to the outlet. This can be seen as a width function based unit hydrograph without diffusion, and goes back at least 20 years (e.g. Franchini and O'Connell, 1996). The distance to the outlet is combined with the travel speed parameter ($\tau$) to determine the lag time for each pixel (see the green line and shifting of columns in Fig. 1). An example of this concept is presented in Fig. 4a.

The width function in Fig. 4a reflects the shape and stream network of a synthetic catchment. The distance of each pixel can be translated to a time delay using the travel speed parameter $\tau$. This delay is in discrete steps (see Fig. 4b), determined by the time step of the model. Slower flow speeds result in larger time delays, and vice versa. The hydrograph in Fig. 4c shows that peaks are more attenuated with slower flow speeds. In this example, we assume homogeneous precipitation across the entire

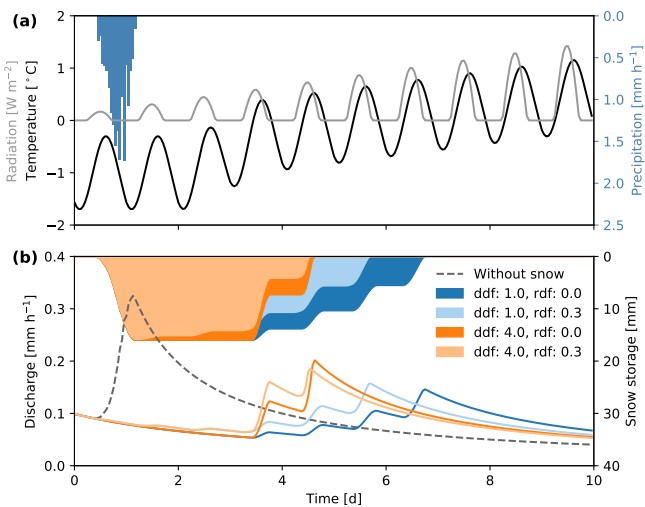

**Figure 3.** Simulation results with and without the snow conceptualization. Top panel shows the model input, bottom panel shows the model output. The four hydrographs represent runs with different snow parameter values (see legend, in mm h$^{-1}$). We fixed the critical temperature $T_0$ to $0°$C. To visualize the influence of the different parameters, we used unrealistically high $ddf$ and $rdf$ values.

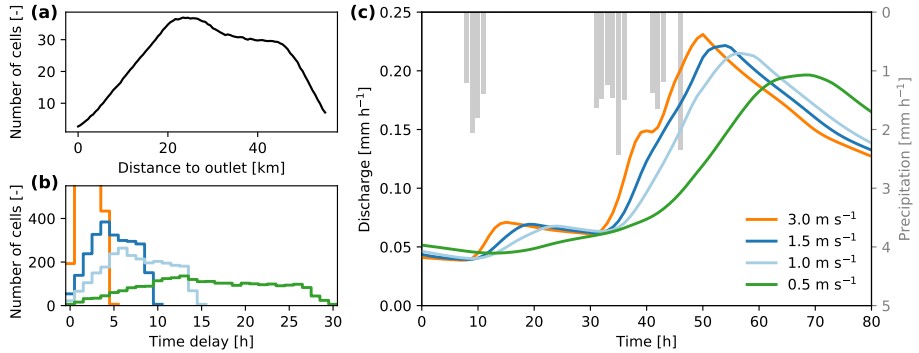

**Figure 4.** The routing concept visualized: a) the width function of a synthetic catchment, b) the corresponding travel speed given four different $\tau$ values, and c) the effect of different $\tau$ values on the hydrograph.

catchment, but in reality, the heterogeneity of precipitation events also influences how the discharge peak is attenuated. This concept does not include diffusion of flood waves, but only incorporates advection.

## 3 Technical aspects

### 3.1 Model implementation

One of the main aims of this model was computational efficiency. However, we do not claim that we have built the most computationally efficient conceptual hydrological model, since we sacrificed some calculation time for the user-friendly and hugely popular Python programming language. By choosing this language, we encourage users of the model to adapt, improve and change all components of the model, in order to find answers to their research questions. The model is written in Python 3.6 and largely uses the Numpy library. Numpy uses C libraries to ensure fast computations over entire arrays, something the base

functions of Python do not offer. Due to the simplicity of the simple dynamical systems approach, we need to solve only one equation (besides the snow conceptualization). Numpy allows functions to be vectorized: receiving and outputting an array of values, while applying the same function to each individual value. This is computationally more efficient than the step–by–step application of the same function to each element in the array.

  To get an idea of the computational efficiency of the model, example run times are presented in Fig. 5. We ran the model

for 3 months at an hourly time step using synthetic forcing data for a wide range of model cells, reflecting different catchment sizes or model resolutions, to get an idea how the computational demand scales with catchment size or resolution. We separated the time spent in the numerical solver, the IO operations, and the routing module. In the IO operations, we include reading the settings file, reading the input data, and writing the model results. The grey bars in Fig. 5 represent the "traditional" modelling approach, where the numerical solver is called for each individual pixel and time step. This extra for–loop drastically increases

runtimes by a factor of more than 10 (this is also a property of Python, being an interpreted language instead of a compiled language). To demonstrate the enormous potential of this model to explore uncertainty and spatial patterns in parameters: simulating Europe for three months at hourly time steps and at a resolution of $5 \times 5$ km$^2$ (roughly $2^{18.6}$ pixels) would take approximately 5 minutes with the efficient dS2 model. The runtimes show a small inconsistency in the increase of runtimes with the increase in pixels. This is most likely the result of an internal Python or Numpy threshold. The current version of dS2

does not yet support automatic multi-threading, which is expected to even further reduce runtimes, especially in large basins.

  When applying the model to very large basins and/or running for very long periods, the random access memory (RAM) can become a limiting factor, as storing data in the RAM is by far most efficient. To prevent exceeding the RAM, the user can define the maximum amount of RAM the model is allowed to use, and dS2 will chunk the input and output data accordingly. For example, running the model for a basin with 1500 pixels for a period of 4 years on a hourly time step would require

approximately 800MB. Additionally, to reduce the time spent with reading and writing the files on the disk, we rely on a special data format: Numpy memmap. This data file directly maps arrays to the hard drive without storing any metadata, to ensure fast reading and writing. The downside is that this format does not have any metadata, meaning the shape and data

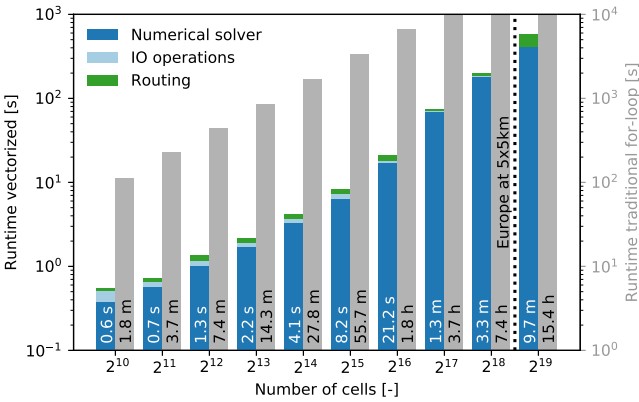

**Figure 5.** Runtimes for a simulation of three months with an hourly time step with varying number of pixels, ran on a single core of a normal desktop (Intel Core i7-6700, 16GB RAM). The grey bars represent model run times where the numerical solver is called for each individual pixel, and are keyed to the right hand scale, making the difference in runtimes appear smaller than it really is. The dotted line represents the number of pixels when simulating Europe at $5{\times}5$ km$^2$ resolution.

type needs to be known prior to reading the file. In order to store the model output in a more common format, the model can transform the Numpy memmap data to the more commonly used NetCDF format, including meta–data.

### 3.2 Adaptive time stepping

When solving a differential equation, one would ideally opt for an analytical solution. However, many equations (including Eq.
3) do not have an analytical solution, so one needs to use numerical solvers to solve the differential equation. The downside of numerical methods is that they might introduce numerical errors and/or numerical instability, as they are approximations of the analytical solution. Many different numerical methods exist, varying in complexity and numerical accuracy. The importance and potential problems of numerical solutions have been described and studied extensively (Clark and Kavetski, 2010; Kavetski and Clark, 2010, 2011). For example, the most simple method is the first order explicit Euler scheme, meaning that the value
at $t_{n+1}$ is based on the values given at $t_n$. This method is the simplest approximation of the value at $t_{n+1}$, but is as a first order method also prone to introduce large numerical errors. A more robust and popular method is the explicit Runge–Kutta 4 scheme, which uses four estimations (fourth order) between $t_t$ and $t_{n+1}$ to give a more accurate estimation of the value at $t_{n+1}$. Furthermore, using higher order (or implicit) solvers reduces the risk of numerical instability. Many more methods are available, which are higher order and/or use different ways to calculate the "intermediate" steps. Numerical solvers in most
hydrological models are rather basic, although it is known that the numerical solver can significantly influence the model output (Schoups et al., 2010). In particular in strongly non-linear problems, a robust numerical solver is important.

Since the simple dynamical systems approach allows for non-linear reservoirs, the simulated response can vary over several orders of magnitude within a single time step. This non-linearity can cause numerical errors, which can be prevented by

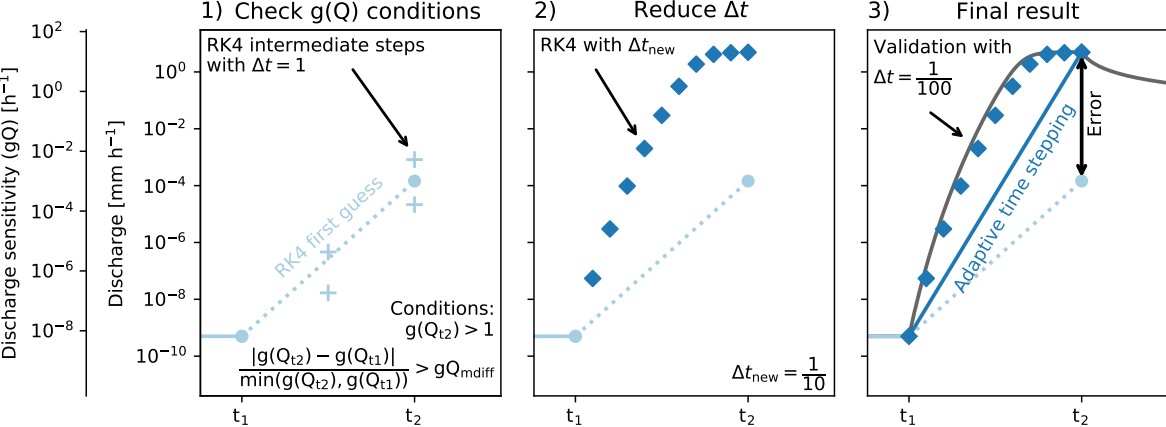

**Figure 6.** Three steps in the adaptive time stepping scheme. Note that in extreme cases, both $Q$ and $g(Q)$ can change over several orders of magnitude between $t_1$ and $t_2$, as can be seen from both y-axes. The plus symbols in step 1 indicate the intermediate values by the RK4 solver, which combine into the blue circle at $t_2$. The blue diamonds in step 2 show the intermediate steps calculated with RK4 but at a smaller timestep. Only the value at $t_2$ is stored. The error in step 3 represents the numerical error without the adaptive time stepping scheme.

reducing the time step. Several options are available, where the Cash–Karp method is the typical textbook approach to explicitly solving differential equations (Cash and Karp, 1990). This method is based on the Runge–Kutta scheme, and uses the difference between fourth and fifth order estimates to measure the potential numerical error. This can be used to reduce the time step if the difference exceeds a certain threshold. However, to ensure optimal computational efficiency, we decided to incorporate the

knowledge about the differential equation (Eq. (3)) to determine whether time step reduction is necessary. Therefore, we have implemented the explicit fourth order Runge–Kutta (RK4) scheme with an adaptive time stepping scheme. We have identified cases where the RK4 scheme can become numerically unstable, and used these cases to reduce the internal time step. These cases are explained in more detail below. This adaptive time stepping scheme is visualized in Fig. 6.

The adaptive time stepping scheme operates in three steps. In the first step, the discharge for the next time step is calculated

using the explicit Runge–Kutta 4 scheme. For both time steps, the discharge sensitivities are calculated based on the following equation:

$$gQ_{\mathrm{diff}} = \frac{|g(Q)_{t2} - g(Q)_{t1}|}{\min(g(Q)_{t2} - g(Q)_{t1})}, \tag{14}$$

where the numerator represents the absolute difference between the two discharge sensitivities, and the denominator represents the lowest discharge sensitivity to ensure the comparison also works during the falling limb of the hydrograph. The time step

will be reduced whenever at least one of two conditions is met: (1) when $g(Q_{t2}) \cdot \Delta t > 1$, or (2) when $gQ_{\mathrm{diff}}$ exceeds a certain threshold ($gQ_{\mathrm{mdiff}}$). The first condition requires a smaller time step, since values of $g(Q) \cdot \Delta t > 1$ indicate that the system is extremely sensitive, and that a smaller time step is required to optimally account for this sensitivity. For this condition, the time

step is reduced according to the following equation:

$$\frac{\Delta t}{\Delta t_{\text{new}}} = \max(5, \min(g(Q_{\text{t2}}) \cdot 10, 50)), \tag{15}$$

where the reduction in $\Delta t$ is dependent on the size of $g(Q_{\text{t2}})$. We defined a lower and upper limit to the time step reduction, to ensure that the solver gains enough precision but does not spend too much time on a single time step. Both upper and lower limits can be changed by the user. The second condition is triggered when $g(Q)$ covers several orders of magnitude in a single time step. A threshold $gQ_{\text{diff}}$ is defined, describing how many times $g(Q_{\text{t2}})$ is allowed to deviate from $g(Q_{\text{t1}})$. If this threshold is exceeded ($gQ_{\text{diff}} > gQ_{\text{mdiff}}$), a smaller time step is required since RK4 is not able to accurately solve the differential equation over this many orders of magnitude. The resulting reduced time step is based on the following equation:

$$\frac{\Delta t}{\Delta t_{\text{new}}} = \max(5, \min(gQ_{\text{diff}}^{\text{dt\_reduction}}, 50)), \tag{16}$$

where the reduction in $\Delta t$ is dependent on the difference between $g(Q_{\text{t1}})$ and $g(Q_{\text{t2}})$, raised to the power $\text{dt\_reduction}$, and the same lower and upper limits to the time step reduction are used. The RK4 scheme is used over these reduced time steps, until the original time step ($t_2$) is reached. This final value is saved as output, and the intermediate steps are discarded. In Fig. 6 step 3, we validate the adaptive time stepping scheme with an even finer time step ($\frac{1}{100}$) which we assume approaches the analytical solution, and we conclude that the adaptive time stepping scheme yields reliable results. This figure shows that the adaptive time stepping scheme avoided a large numerical error (the difference between the two solutions: $Q_{t2} \approx 10^0$ mm h$^{-1}$ vs $Q_{t2} \approx 10^{-4}$ mm h$^{-1}$).

During periods with low discharge and relatively high evaporation, the numerical solver can result in negative discharge amounts. To prevent this from happening, we define a threshold for discharge (which thus corresponds to a threshold storage level), below which no evaporation is allowed to occur. This mimics evaporation reduction during periods with low storage volumes, as discharge and storage are directly linked. For the pixels where the discharge is below this threshold, the evaporation rate is set to 0 mm h$^{-1}$. The value of the discharge threshold is currently set at $10^{-4}$ mm h$^{-1}$, but can be altered by the user. All model parameters are presented in Table A1.

### 3.3 Closure of the water balance

The concept of this model is based on the water balance, yet it does not explicitly solve the water balance as most hydrological models do. The water balance is indirectly solved by calculating changes in storage. To check whether the model still respects the water balance, the closure of the water balance is calculated using the following equation:

$$\phi = P_{\text{t}} - E_{\text{t}} - \int_{t-1}^{t} Q - \Delta S, \tag{17}$$

$$\Delta S = S(Q_{\text{t}}) - S(Q_{\text{t}-1}), \tag{18}$$

where $\phi$ is the error in the water balance for each time step $t$, which ideally should result in a value of zero. The change in storage can be calculated using the storage at the beginning and at the end of the time step. The integral over $Q$ indicates

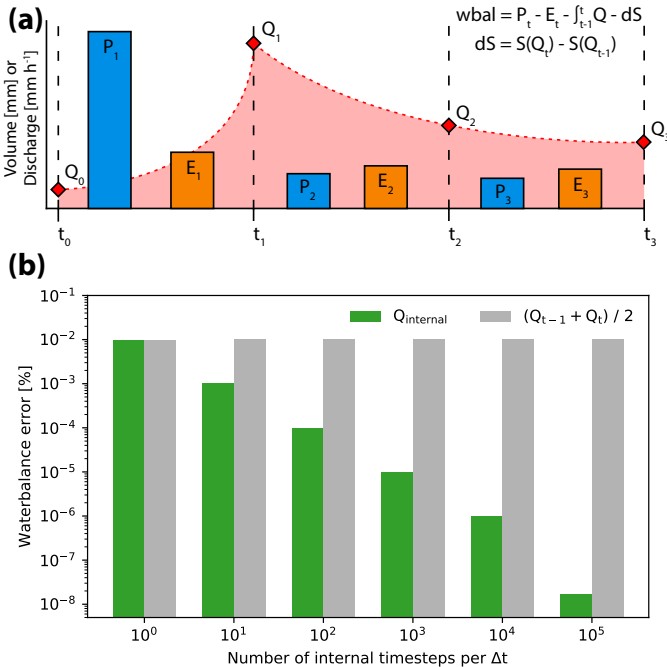

**Figure 7.** Closure of the water balance per time step. Panel (a) shows the timing of the different variables. Red area under the dotted line indicates the total volume of water discharged per time step. Panel (b) shows the average absolute error in the water balance when the volume of water is estimated with smaller internal time steps ($Q_{internal}$), where the error is defined as the percentage of the total precipitation during the simulation period.

that one needs to consider the volume of water that is discharged during the time step. This depends on how the variables are considered in time within the model. Figure 7a explains how the different fluxes are positioned within the model. The model assumes that the precipitation and evaporation values are summations of the entire duration of a single time step. The resulting discharge values are, however, only representative at the end of the time step, and not a summation over the time step. Due to the strong non-linearity of the system, the total volume of water discharged during a single time step cannot be represented by the discharge at the end of the time step, as indicated by the dotted red line in Fig. 7a.

To estimate the volume of water discharged during the time step, one needs a discharge value that is representative for the volume of water discharged during that time step. The easiest way to calculate this is to take the average of the discharges at time steps $t$ and $t-1$. However, if the sensitivity function is strongly non-linear, this average might not be representative for the volume of water discharged during that time step. To improve the volume estimation, one can subdivide each time step, and base the volume estimation on the mean discharge of the resulting shorter steps ($Q_{internal}$). This comparison is shown in Fig 7b. The grey bars represent the error in the water balance when the total discharge volume is estimated using the discharge averaged over time steps $t$ and $t-1$. This error does not reduce with an increase in internal time steps, as the discharge prediction at $t$ changes only marginally. However, if the total discharge volume is estimated using the discharge at each internal

time step, the total error in the water balance is reduced from $10^{-2}\%$ to $10^{-8}\%$. This indicates that the model concept is able to successfully close the water balance, given that the discharge volume per time step is accurately accounted for. Please note that this example is to show that the concept successfully closes the water balance at each time step. The current version of the model only outputs discharge at the end of the time step, as calculating $Q_{\mathrm{internal}}$ does not add to the numerical accuracy of the discharge calculation at the end of the timestep.

## 4  Parameter sensitivity

We investigated the response of the model to the five main parameters ($\alpha$, $\beta$, $\gamma$, $\epsilon$, $\tau$; excluding the snow parameters) by plotting the response surface over realistic parameter ranges. These ranges are based on comparing the shape of the resulting discharge sensitivity function with the previous studies that use the SDS approach. The results of this analysis are presented in Fig. 8. We selected the Kling–Gupta efficiency (KGE) as the performance metric. We created a synthetic time series of observations based on parameters that are in the middle of each subplot, which are used to calculate the KGE. It is striking that the three discharge sensitivity parameters ($\alpha$, $\beta$ and $\gamma$) show large regions with similar model performance (the dark blue regions), and seem to be correlated, which Melsen et al. (2014) also showed for the $\alpha$ and $\beta$ parameters for a lumped application. However, based on theoretical considerations (Troch et al., 1993; Brutsaert and Lopez, 1998; Tague and Grant, 2004; Rupp and Selker, 2006; Lyon and Troch, 2007; Rupp and Woods, 2008, e.g.), we can conclude that at least the $\alpha$ and $\beta$ parameters are required to optimally capture the discharge sensitivity, and based on other studies using the simple dynamical systems approach (Kirchner, 2009; Teuling et al., 2010; Krier et al., 2012; Adamovic et al., 2015), that a third parameter ($\gamma$ in this case) is required to capture the curvature in the $g(Q)$ relation. These response surface graphs indicate that local optimization algorithms might struggle to find the global maximum due to the large equifinality regions. We therefore recommend not to use a single parameter combination, but rather to use multiple parameter sets to account for equifinality.

Furthermore, we also analysed the parameter sensitivity according to Sobol' (2001), Saltelli (2002), and Saltelli et al. (2010). The global sensitivity analysis method is designed to analyse the sensitivity of different performance statistics to each parameter. Sobol' sensitivity analysis works optimally in a case where parameters are not correlated. Even though this condition is not fully met in our model application, we use Sobol' because it is the most widely used method to investigate parameter sensitivity. Furthermore, the aim of this sensitivity analysis is to give a first impression of the influence of the parameters on performance metrics. To perform this analysis, a set of $n \cdot (2k+2)$ parameter combinations is required, where $k$ is the number of parameters, and $n$ is the number of samples that sample the parameter space. We chose a sample of $1,000$, and focused on the five main parameters ($\alpha$, $\beta$, $\gamma$, $\epsilon$, $\tau$, excluding the snow parameters) resulting in $12,000$ parameter sets. Usually, this is a rather computationally expensive method, but due to the computational efficiency of this model, we were able to run all parameter sets in just under 6 hours (model with 100 cells, simulation period of 2 years on hourly time step, on a normal desktop with an Intel Core i7-6700 and 16GB RAM). The parameter boundaries were set to the same values as used in the response surface plots. For each parameter combination, multiple performance statistics were calculated. Next, Sobol' sensitivity analysis is able to extract the influence of each parameter on the variation in the performance statistic caused only by that parameter ("main

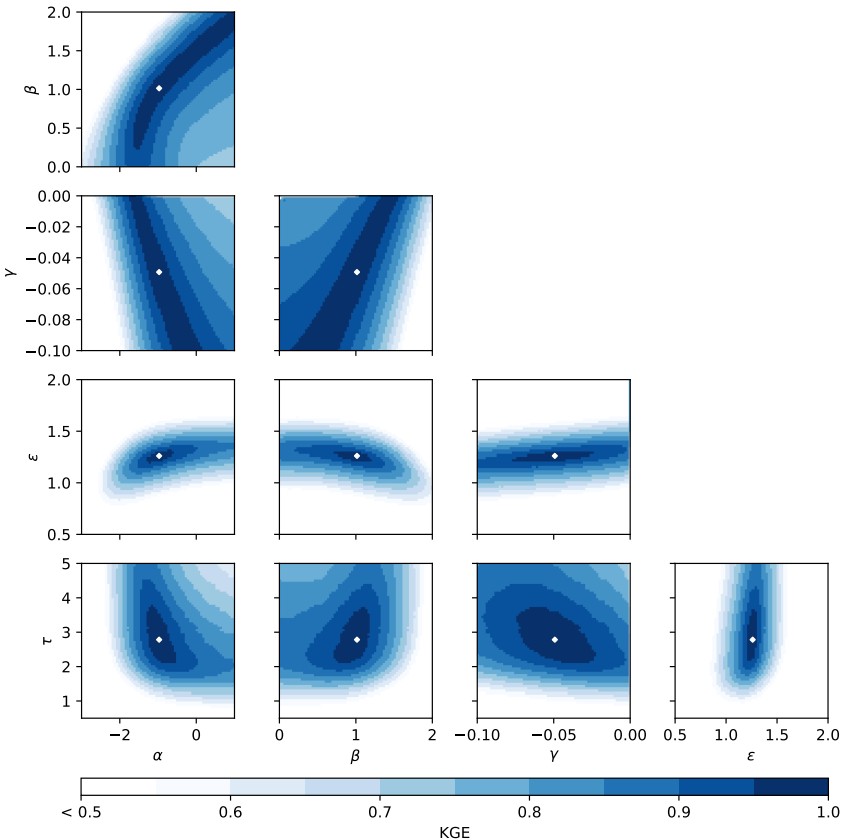

**Figure 8.** Response surface plots for all main parameter combinations with the Kling–Gupta efficiency (KGE) as the performance metric for a synthetic experiment. The white dot in the middle of each graph represents the location where KGE is equal to 1. Each plot consists of 4900 model runs — where the model has 200 cells, and was run for one year on hourly timestep — and took 1 hour and 15 minutes to run on a normal desktop (Intel Core i7-6700, 16GB RAM).

effect"), and the influence of the parameter including all variance caused by interactions with other parameters ("total effect") (Sobol′, 2001; Saltelli, 2002; Saltelli et al., 2010). The results from the analysis are presented in Fig. 9.

It is clearly visible that parameter sensitivity depends on the performance metric used. The Nash–Sutcliffe Efficiency (NSE) and Kling–Gupta Efficiency (KGE) show roughly the same response, as is expected. Nash–Sutcliffe calculated on the logarithmic discharge values shows a very different response, with $\gamma$ being one of the most sensitive parameters. This is in line with our expectations, as the $\gamma$ parameter describes the downward curvature of the $g(Q)$ function, where it mostly affects the lower discharge volumes. Additionally, this parameter is not very prominently visible in the NSE and KGE sensitivity plots, since these metrics tend to put more focus on higher discharges. These graphs also show that there are some parameter interactions influencing the results, indicated by the difference between the main and total effects. This was already visible in Fig. 8, where

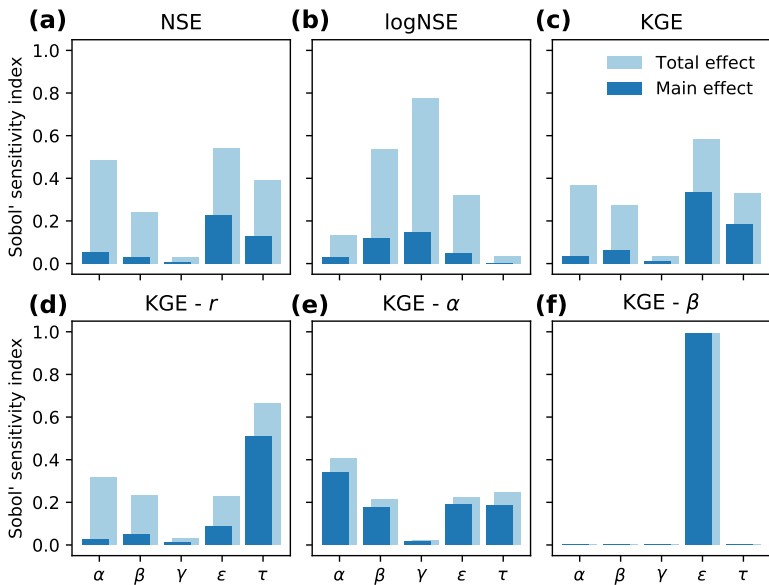

**Figure 9.** Sobol' parameter sensitivity for a synthetic experiment, shown for different performance statistics. The main effect presents the sensitivity induced by that parameter alone, and the total effect quantifies the sensitivity when parameter interactions are taken into account as well. The bottom row (panels d to f) shows the three components of the Kling–Gupta Efficiency: Pearson correlation coefficient (KGE - $r$), ratio of variability (KGE - $\alpha$) and bias (KGE - $\beta$).

we see correlations between the three $g(Q)$ parameters. Since this analysis is performed to give a first evaluation of how the parameters affect model output, we will focus on the main effect.

The most interesting patterns are visible in the bottom three subplots, where the three components of the KGE are presented. The correlation coefficient (KGE - $r$, Fig. 9d) is most sensitive for the routing parameter $\tau$, since this parameter essentially deals with the timing of discharge peaks. The ratio of variability (KGE - $\alpha$, Fig. 9e) does not show any individual parameter to be most important, yet all parameters show some degree of sensitivity. For the bias (KGE - $\beta$, Fig. 9f), the evaporation parameter $\epsilon$ is by far the most sensitive parameter, since it determines the total volume of water that evaporates and thus the total volume of discharge. All other parameters can only control either the response of discharge to precipitation or the timing of the peaks. Since the roles of the parameters $\epsilon$ and $\tau$ are relatively clearly defined, predominantly affecting the bias and correlation, respectively, we hypothesize that these roles can be used for more efficient optimization procedures by reducing the number of dimensions. One could optimize the $\epsilon$ parameter only on the bias of the simulations, followed by an optimization of the $\tau$ parameter on the correlation. Finally, the three $g(Q)$ parameters can be optimized on either the ratio of variability or the total Kling–Gupta efficiency. This reduces the optimization from a 5-dimensional problem to two 1-dimensional problems and a single 3-dimensional problem.

## 5  Example application

In this section, we apply the model to the Thur, a mesoscale basin (1700 km$^2$) in the Swiss Alps. This basin was selected since it contains many measurement locations, where the catchment used by Teuling et al. (2010) is one of the sub-basins. Since Teuling et al. (2010) showed that the SDS concept is able to simulate the discharge at small scale, we take this opportunity to see how the model performs at different spatial extents. The model was applied at a 1 km$^2$ resolution, using distributed forcing. We used the same forcing data as Melsen et al. (2016a), where the data is interpolated using the pre-processing tool WINMET of the PREVAH modelling system (Viviroli et al., 2009; Fundel and Zappa, 2011). For a more detailed explanation of the basin and the data used, we refer to Melsen et al. (2016a).

For this application, we focus on three sub-basins in the Thur to validate the model at three different spatial extents (see Fig 10a). We used a Monte Carlo approach to generate 25,000 parameter sets to ensure good coverage in the parameter space. The parameter boundaries are based on the ranges used in Fig. 8. We based the snow parameters on the study by Teuling et al. (2010). To capture the spread in parameters as a result of equifinality, the 100 best runs based on the Kling–Gupta Efficiency (KGE) are selected for each sub-basin. The resulting discharge time series are presented in Fig. 10b to Fig. 10d.

In Fig. 10b to Fig. 10d the simulated discharge is compared with the observed discharge for three different basins. The three basins are ordered from small to large. The simple dynamical systems approach has already been applied to the Rietholzbach, the smallest of the three basins in Fig. 10, by Teuling et al. (2010), but in a lumped fashion. We see that the distributed version of this concept is able to correctly simulate the discharge in this basin, but also in the bigger basins. In the top left corner of each subplot, 4 performance statistics are presented based on the run with the highest Kling–Gupta Efficiency: the Kling–Gupta Efficiency (maxKGE), Pearson correlation coefficient ($r$), ratio of variability ($\alpha$), and bias ($\beta$). The KGE value within brackets represents the Kling–Gupta Efficiency without the routing module. These metrics show that there is no substantial decrease in model performance with increasing catchment size. This indicates that the model is able to correctly simulate the discharge, independent of the catchment size. We do see, however, a clear decrease in the KGE values when no temporal delay is added via the routing scheme. In the smallest basin, both KGE values show the same result, yet in the largest basin, we see a substantial decrease in model performance. This indicates, in line with expectations, that the routing module is required to optimally simulate larger basins.

To elaborate this, we zoom in to a single event in Fig. 10e. This graph shows the discharge response of the three sub-basins, and the effect of the routing module. For this event, we selected the one parameter set from the 100 sets that performed best over the period depicted in panels (b) to (d). We see that, although the magnitude of the peak is not fully captured in the Rietholzbach basin, the timing of the peak is well simulated. Since this is a very small basin, the time lag induced by the routing module is less than one hour, meaning that the run without the routing module gave the same result. At Mogelsberg, both the timing and magnitude of the discharge peak are well simulated by dS2. The routing module adds a correct temporal delay to the discharge peak. At the main catchment outlet (Andelfingen), we again see that the model correctly simulates the timing of the discharge, but struggles to reach the correct magnitude. This is likely related to errors in the precipitation product rather than the model structure, as dS2 struggled to reach good performance in two basins in the north western part of the

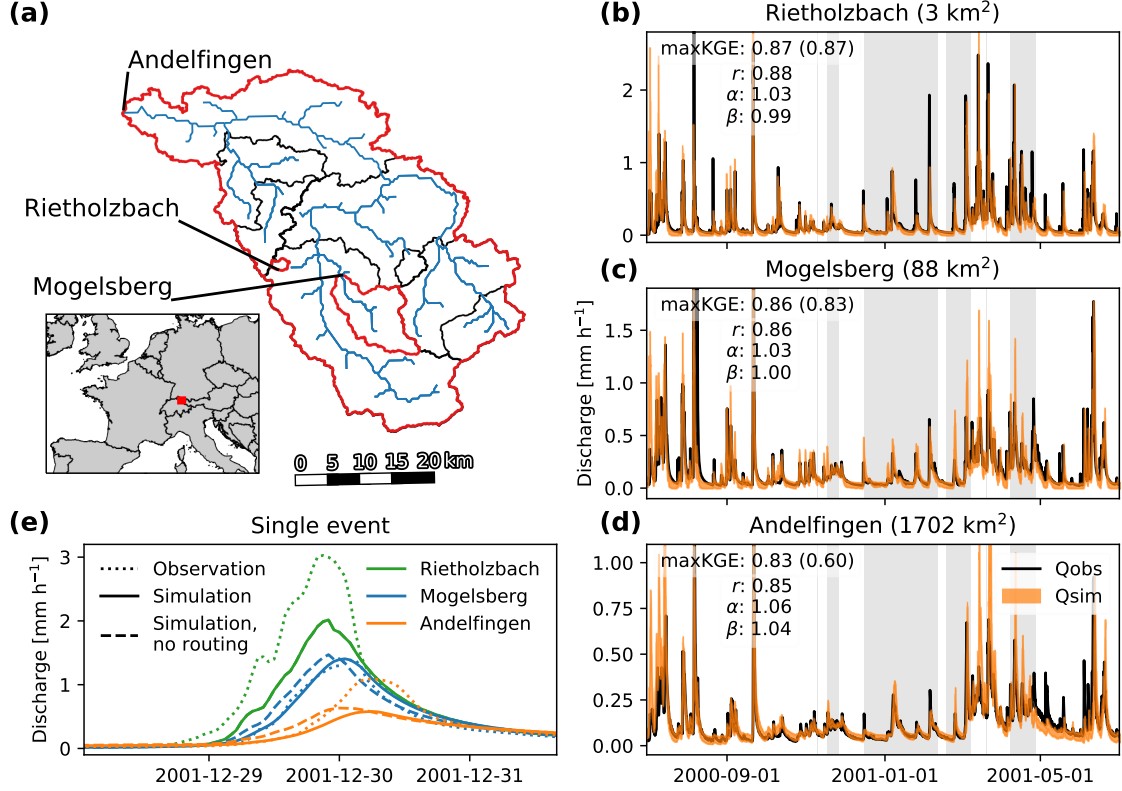

**Figure 10.** Application of dS2 to the Thur basin in Switzerland. Panel (a) depicts the stream network and the catchments used in this example are highlighted in red: Rietholzbach, Mogelsberg and Andelfingen. Panels (b) to (d) present the model output in the three basins, covering different orders of catchment size. The shaded orange region shows the minimum and maximum discharge of the 100 best runs out of $25,000$, selected based on KGE value. Periods where more than 20% of the entire basin is covered with snow are indicated with the grey background. Performance values show the statistics for the best model run, with the value between brackets showing the model performance when the routing module is disabled. Graph (e) shows how a single precipitation event is translated through the different (sub-)basins, where the dotted line indicates the observed discharge, the solid line represents the simulated discharge, and the dashed line represents the simulated discharge without a temporal delay induced by the routing module. Note that the simulation result without routing for Rietholzbach overlaps the simulation result with routing.

catchment (Frauenfeld and Wängi). This, together with the too low discharge peak in the Rietholzbach basin, might explain the lack of discharge passing Andelfingen. Similar to Mogelsberg, the routing module does add a correct filtering of the discharge signal due to the distribution of the stream network. Overall, the runs without the routing module indicate that the impact of the routing module depends on the size of the catchment, where the difference between these runs is largest in the biggest

catchment. The differences between these runs explain the reduction in model performance in Fig. 10b to Fig. 10d, measured using the KGE without routing.

## 6 Discussion

As stated in the introduction, the aim of this paper is not to present the best or most comprehensive conceptual rainfall–runoff model, but to develop a model that can more easily be used to perform studies that require a large number of runs, such as

uncertainty or sensitivity studies. As a result, this model concept has several limitations, either linked to the original simple dynamical systems approach or linked to the (distributed) implementation of this concept.

A limitation of the simple dynamical systems approach is that it assumes that the storage–discharge relation is the same on the rising limb and on the falling limb of the hydrograph. Studies have shown that hysteresis exists in multiple basins, especially those dominated by a variable contributing area (Spence et al., 2010; Fovet et al., 2015). However, most common

hydrological models do not explicitly account for hysteresis, and often struggle to correctly simulate the dry–to–wet transitions (de Boer-Euser et al., 2017). In our study, the simple dynamical systems approach assumes a fixed storage–discharge relation, with discharge being a function only of storage. In earlier studies where the concept was applied in a lumped fashion, this did not limit the performance of the concept, especially in the studies where the $g(Q)$ parameters were retrieved from the discharge observations (Kirchner, 2009; Teuling et al., 2010). In our case study, this also did not seem to be a limiting factor.

We do expect the model performance to be affected in basins exhibiting a strong hysteresis, yet we expect that one might define the $g(Q)$ function in a way that best captures the average behaviour of the rising and falling limbs.

Surface runoff as result of either infiltration excess or saturation excess is not explicitly accounted for. Surface runoff induced by oversaturated soils is implicitly accounted for in the discharge sensitivity function, which will reach high sensitivity values with high discharge (i.e. storage). During precipitation events with high intensity, surface runoff can also occur when the soil's

infiltration capacity cannot accommodate the precipitation intensity. The model might therefore underestimate some discharge peaks resulting from infiltration excess overland flow. However, we expect that the influence of this process is minor in humid catchments. Not accounting for this process relates back to the purpose of this model, where we do not focus on correctly representing catchments' internal fluxes, but instead focus on simulating the total discharge in a computationally efficient manner. It does, however, disqualify this concept for catchments with extreme precipitation events and the corresponding

overland flow processes.

Furthermore, this model relies on the user to provide accurate actual evapotranspiration values as input for the model. As can be seen in Eq. (3), evaporation is directly taken into account in determining the change in storage. Due to the simplicity of the concept, the model does not contain any evaporation reduction processes as result of e.g. soil moisture stress. The model

disables evaporation when discharge drops below a threshold, behaving like an on–off switch, which is not how it is observed in reality. This reduction is most importantly preventing the model from calculating negative discharge values, while trying to behave like a real process.

The vectorized implementation limits the ability to transfer water between neighbouring pixels. This subsurface lateral flow is mostly driven by gravity, and is therefore expected to be most important in regions with large elevation differences. However, at the current recommend spatial–temporal scale of application ($\sim 1$ km$^2$ and 1 h), we expect lateral subsurface flow to be relatively unimportant. At smaller spatial scales or longer time scales, flow between pixels can become more important. To stay in line with the original scale of the simple dynamical systems approach, we do not recommend to further reduce the spatial resolution or to further increase the temporal resolution. Furthermore, most common hydrological models also do not account for flow between pixels (Liang et al., 1996; Arnold et al., 1998; Terink et al., 2015).

As proposed by Kirchner (2009), this concept can be used to "do hydrology backwards": infer precipitation from the discharge time series. Unfortunately, the distributed implementation of this concept complicates the use of this model to infer spatially distributed rainfall maps. Since the resulting discharge is an average of the integrated catchment response, it is impossible to infer the location of the precipitation event. This will become especially difficult in large basins, where the time lags induced by routing will become larger. A study by Pan and Wood (2013) describes a method to infer precipitation from streamflow observations, though it requires a relatively large number of observations. This method can potentially be applied to the model presented in this paper.

Finally, the current technical implementation of the model implies two other limitations. First, the routing scheme only induces a time lag on the discharge peak, and does not account for diffusion of the wave as it travels through the river network. For smaller basins, the effect of diffusion is only minor, but when simulating larger basins such as the Rhine, diffusion of the discharge wave will play a more important role. We recommend to use a more advanced routing scheme when applying the model to larger basins. The model can output a NetCDF file with runoff generated in each pixel per time step, which can easily be connected to other routing software such as SOBEK. Secondly, the current version of the model does not (yet) support multi-threading, meaning that the model currently only runs on a single thread. Spreading the computational load over multiple threads is expected to further reduce the time required to run the model, especially in large basins. To make the model more efficient in single thread mode, one can split the entire basin into several smaller sections and simultaneously run the model for each section. After the simulation is complete, the user can combine the several resulting pixel-wise runoff maps, and apply a routing scheme to the entire basin.

## 7  Conclusions

The distributed simple dynamical systems (dS2) model is a new rainfall–runoff model that aims to simulate discharge in a computationally efficient manner. This model is based on the simple dynamical systems approach proposed by Kirchner (2009), but has been modified to be applied in a distributed fashion, written in Python. In this way, the concept can be applied to larger basins, while respecting the original spatial and temporal scale of the concept, and also make use of high–resolution

data. We have extended the concept with a snow module and a simple routing module. The snow module can be turned on or off, depending on the basin of interest, and the routing module is required to transport water through the basin towards the outlet. The flexibility in the discharge sensitivity ($g(Q)$) function and the resulting strong non-linearity make this model different from more common "bucket-type" models. As a result, dS2 is able to quickly simulate hydrological response at relatively high

spatial ($\sim$1 km$^2$) and temporal ($\sim$1 hour) resolution.

    Synthetic examples demonstrate that, although dS2 solves the water balance implicitly, the model is able to accurately close the water balance. The response surface plots for all parameter combinations show that there are some correlations between the parameters, especially for the three discharge sensitivity parameters. However, we also showed that the parameters clearly influence different performance metrics, which provides the opportunity to reduce the calculation time of optimization

algorithms. Finally, we have applied the model in the Thur basin as a case study to validate the performance of the model. Our model is able to correctly simulate discharge, both at the local scale (e.g. the Rietholzbach catchment in Switzerland, 3 km$^2$) and at the mesoscale (entire Thur basin in Switzerland, 1700 km$^2$), without a decrease in model performance as catchment size increases.

    The distributed simple dynamical systems (dS2) model has several unique strengths compared to other rainfall–runoff mod-

els: (1) it is computationally efficient even at high temporal and spatial resolutions, (2) it only has five parameters to calibrate, (3) two parameters have a clearly defined influence on the discharge time series making them easy to identify, and (4) the Python model code is open source and easily adjustable. The computational efficiency of this model creates the opportunity to answer different research questions. Since the model is able to simulate regions in relatively short amounts of time, performing sensitivity and uncertainty analyses with a large number of samples becomes feasible. Since the model is distributed, the

sensitivity and uncertainty analyses can be performed both on a spatial and temporal basis. Furthermore, this model can be a valuable tool for educational purposes, to explain and directly show the effects of modifying parameters in distinct groups (e.g. hydrological response, snow, routing, evaporation). Overall, this makes dS2 a valuable addition to the already large pool of conceptual rainfall–runoff models, both for researchers and for practitioners interested in large sample studies.

*Code availability.* The model code is open-source and is archived at the 4TU repository: https://doi.org/10.4121/uuid:cc8e0008-ab1f-43ee-b50d-24de01d2

(Buitink et al., 2019). The latest version of the dS2 model can be found on GitHub: https://github.com/JoostBuitink/dS2 (last access: 9 January 2020). On the same GitHub repository, a brief manual can be found which briefly explains what input data is required and how to set up and run the model. This manual uses the Rhine basin in Europe with the ERA5 dataset as an example.

*Author contributions.* JB, LAM and AJT conceived the idea behind dS2. JB developed the dS2 model, with valuable inputs from all co-authors. JB performed the synthetic experiments and case study, LAM and AJT helped with interpreting the results. JB prepared the

manuscript, with significant contributions from all co-authors.

**Table A1.** List of dS2 parameters. All parameters after rdf only influence either the numerical stability and/or the IO operations of the model, not the model output.

| Name | Description | Example value | Unit |
|---|---|---:|---|
| $\alpha$ | gQ parameter - intersect | -2.5 | - |
| $\beta$ | gQ parameter - slope | 0.85 | - |
| $\gamma$ | gQ parameter - curvature | -0.010 | - |
| $\epsilon$ | Reduction factor for evaporation | 0.89 | - |
| $\tau$ | Travel speed of water through the catchment | 2 | $\mathrm{m\,s^{-1}}$ |
| T0 | Critical temperature | 0 | °C |
| ddf | Degree day factor for snowmelt | 2 | $\mathrm{mm\,day^{-1}\,°C^{-1}}$ |
| rdf | Conversion factor for energy to snowmelt | 0.26 | $\mathrm{mm\,day^{-1}\,(W/m^2)^{-1}}$ |
| Qt | Threshold below which evaporation is set to zero | $10^{-4}$ | $\mathrm{mm\,h^{-1}}$ |
| dt | Size of time step ($\Delta t$) | 1 | h |
| max_RAM | Maximum RAM the model is allowed to use | 1024 | MB |
| size_one_value | Size of a single value | 4 | bytes |
| LB | Factor controlling lower boundary for solver, where $Q_t \leq Q_{t-1} \cdot \mathrm{LB}$ | $10^{-4}$ | - |
| max_gQ_difference | Maximum allowed difference in gQ values | 2 | - |
| dt_reduction | Factor controlling the $\Delta t$ reduction | 0.15 | - |
| min_extra_dt | Number of minimum extra $\Delta t$ per $\Delta t$ | 5 | - |
| max_extra_dt | Number of maximum extra $\Delta t$ per $\Delta t$ | 50 | - |

*Acknowledgements.* We would like to thank MeteoSwiss and Massimiliano Zappa for providing the forcing data for the case study.

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
