# Peer review of "A distributed simple dynamical systems approach (dS2 v1.0) for computationally efficient hydrological modelling"

_Geoscientific Model Development, 2019_

## Referee Comment (RC1) · Anonymous Referee #1 · 16 Aug 2019

Dear authors, thank you for submitting your paper. In the following, you will find my comments, suggestions, and opinion of your contribution.

Summary

The paper "A distributed simple dynamical systems approach (dS2 v1.0) for computationally efficient hydrological modelling" is about a distributed hydrological modelling based on the simple dynamical system approach, SDS (Kirchner, 2009). The model is grid-based (cell of 1km2) and enables rainfall-runoff simulations for mesoscale basins (up to 2000 km2) at high temporal and spatial resolution with a new 3-parameter discharge sensitivity function. The model is first developed and assessed for synthetic

experiment and then applied to the alpine Thur catchment (1700 km2). Authors used Monte-Carlo methodology to generate model parameter sets constrained within the synthetic parameter ranges. The model is complemented with snow and routing modules to take into account the lag-time.

General comments

The manuscript is clearly written but has in its present form some major defaults. The authors put forward the novelty of the proposed approach. This is not right since the approach is not unique in its formulation. What is proposed is a simple dynamical system approach by Kirchner applied in its distributed form. Such model, however already exists (please check Adamovic et al. 2016, Journal of Hydrology). The latter was not cited in this article. Adamovic et al. (2016) implemented the SDS approach on Ardeche catchment (around 2400 km2) and developed the model called SIMPLEFLOOD that is computationally fast and coupled with a kinematic wave flow propagation module.

Specific comments

Here are, however, my short comments that I think can contribute to its better content once reformulating and updating the article.

Page 2. Line: 25

The authors mention continental-scale forcing datasets in relation with merged radar data, interpolated station data, and atmospheric reanalysis. It would be valuable for readers if they know in more details about actual spatial but also temporal scales of provided references.

Page 2. Line: 35

The phrase "An efficient distributed conceptual model to tackle these kinds of issues is currently lacking." is not correct. Adamovic et al. (JoH, 2016) already developed this approach at an hourly time step and applied it to the Ardeche catchment.

Page 3. Line: 25

Similar to the previous comment. The approach was already applied to the mesoscale catchment of 2400 km2 (Adamovic et al., 2016).

Page 6. Line: 15

The introduction of a third parameter that avoids having a curvature that diverges computations is interesting even though it has its limitations since it seems that three parameters depend on each other.

Page 8. Flow routing

Authors introduced simplified routing module that is not explicit but rather based on a simple temporal delay to the outlet. More sophisticated flow routing module is described in Adamovic et al. (2016) which takes into account the feedback effects.

Page 12. Closure of the water balance

In the model, the water balance seems closed only in conditions when AET=PET (humid conditions). If the model is to be applied on pan-European scales with different conditions, the question about the AET estimation should be addressed.

Page 13. Parameter sensitivity

It would be useful for a reader to know what the realistic parameter ranges are and how did you choose them since you applied the method on the synthetic catchment in this section.

Page 16. Example application

Line: 23. Did you also run the MC analysis with the fewer or higher number of iterations? Why did you choose 25000? Many authors used fewer iterations for higher-order parametric models (e.g. Tekleab et al. 2011).

Page 17. Line: 7

"Parameter set that performer best for the event". Is it the one that showed the highest KGE? If it is the case, what was the value?

Page 17. Line: 9

"...the timing of the peak is well simulated in all three basins". It seems that for the Andelfingen catchment (Fig. 10e), there is a peak delay of around 6 h between the observation and simulation for this specific event. I would suggest reformulating this sentence and discussing what could be a reason for such peak delay since it is the largest explored catchment. Could it be that the simplified routing module shows the lack of performance in this case? Testing it in more catchments could give a more robust answer.

Technical corrections

Figure 1.

Panels in capitals A, B, C instead of small letters. This will improve the text visibility.

Figure 10 a.

I would suggest colouring the catchments for better visibility.

Figure 10 b, c and d.

I would suggest plotting these figures on a log-scale for better visibility and assessing the influence of evapotranspiration.

Figure 10.e

I would suggest not using the light colours for observations since it is not well readable (i.e. Mogelsberg catchment). Asterix or other dash types could be more appropriate in this case.

References

Adamovic, M., Branger, F., Braud, I., Kralisch, S., 2016. Development of

a data-driven semi-distributed hydrological model for regional scale catchments prone to Mediterranean flash floods. Journal of Hydrology, 541: 173-189. DOI:https://doi.org/10.1016/j.jhydrol.2016.03.032

Tekleab, S., Uhlenbrook, S., Mohamed, Y., Savenije, H. H. G., Temesgen, M., and Wenninger, J.: Water balance modeling of Upper Blue Nile catchments using a top-down approach, Hydrol. Earth Syst. Sci., 15, 2179–2193, doi:10.5194/hess-15-2179-2011, 2011.

---

## Author Comment (AC1) · 22 Aug 2019

Response to Anonymous Referee #1 (AR1)

We thank AR1 for their time and effort to write a constructive comment in order to improve the quality of the manuscript. Below, we respond to the general comment of AR1. We also thank AR1 for the specific comments and technical corrections, which we will clarify/implement in the next version of the manuscript.

We totally agree with the reviewer that a discussion of SIMPLEFLOOD is currently lacking and should be added. However, we do believe that SIMPLEFLOOD and dS2

are different models which complement the currently existing hydrological model selection. Comparing the two models: dS2 is a distributed model on a grid basis, compared to the catchment-based SIMPLEFLOOD model; dS2 has a strong focus on computational efficiency, including a fast but robust numerical solver optimized to solve the SDS approach, compared to the more generic JAMS framework where SIMPLEFLOOD is in embedded; furthermore, the model code of dS2 is publicly available and can easily be modified. In the next version of the manuscript, we will describe the differences between dS2 and SIMPLEFLOOD.

---

## Referee Comment (RC2) · Anonymous Referee #2 · 6 Dec 2019

**1 General comments**

**Summary**   The manuscript *A distributed simple dynamical systems approach (dS2 v1.0) for computationally efficient hydrological modelling* by Buitink and colleagues introduces a distributed hydrological conceptual model oriented at the simple dynamical systems approach by Kirchner (2009). The model has been designed to simulate rainfall–runoff dynamics with high spatial and temporal resolution at small ($10^0$ km$^2$) up to mesoscale ($10^3$ km$^2$) catchments. The model should be straightforward to apply, computationally efficient, and the associated Python code is openly available and easy to understand and modify. In their manuscript the authors present a sensitivity analysis of model parameters and apply the model in the alpine Thur catchment.

Most parts of the paper are well structured and good to understand. However, I see some serious problems with the paper (and the model) that should be resolved before final publication. Even more than for my concerns regarding originality and efficiency this holds true for the applicability of the model, as I will explain in the following paragraphs.

**Originality**   As even the authors admit, many different conceptual hydrological models already exist. Therefore the question arises if we really need yet another model. My personal answer would be: maybe, if we can learn something from it. However, dS2 seems to me rather like a typical black box model and I don't see a way to learn anything about processes or the catchment. Even though the authors see their model as a valuable tool for educational purposes (page 20, line 17 ff.), other conceptual models based on multiple storages (or buckets) are, in my opinion, much more suitable for education. For instance, storages can be related to natural phenomena (e.g. surface, quick subsurface, and baseflow) and students can learn something about processes and parameter sensitivities (for an illustration, see e.g. Fenicia et al., 2016). As far as I can tell, dS2 serves solely as a very simple rainfall–runoff transformation tool and is therefore nothing new. I wonder how we can learn something about processes in a catchment, as the authors claim. To me this is not evident from the example application.

**Efficiency**   The authors stress at many occasions the computational efficiency of their model. On the other hand, computational efficiency has been sacrificed in favour of code readability by implementing it in Python, a widely used scripting language that is known for its well structured syntax in contrast to much more efficient compiled languages such as C++ or Fortran. Therefore, in my view the argument of computational efficiency, which is even advertised in the title of the manuscript, does not hold. A good compromise would have been to outsource the most expensive parts of code to a compiled language and use Python as an interface, as has been done for other models (e.g. TOPMODEL, for which an R interface exists, while the core model is written in Fortran, see Metcalfe, Beven and Freer, 2015).

**Applicability**   I acknowledge that the code of the model is indeed well readable, even for persons with little Python experience (including me). I only wonder how I should apply the model. The authors present an example application in their manuscript but no example data to test the model. What is even more, the input and output file structures are not well explained, neither in their submitted manuscript nor in the assets or the github repository. In that way I don't know how to initialise and apply the model. This is probably the most severe flaw of the presented work. In my opinion it is also violating the core principles of GMD's model code and data policy.

**Sensitivity analysis**   I see a problem with the Sobol' method for variance-based sensitivity analysis that the authors applied. This method may not be applied in case parameters are correlated (which is the case, as is stated in the text). I will provide some more reasoning and suggestions in the specific comments.

**2  Specific comments**

In the following I will provide some specific comments. Quotations from the paper appear as *emphasised text*. Specific passages in the text are referenced as pmLn, meaning page m, line n.

- *"There is a growing need for easy-to-apply models that can utilize the potential of spatially distributed input data"* (p2L26–27). Many (most?) models already utilize spatially distributed information. I also think the demand for *easy-to-apply* models always has been present (whereas the judgement of what is *easy-to-apply* is rather subjective).

- *"water is most often transferred to the outlet as a post-processing function. This is, however, not necessarily the most computationally efficient way to deal with spatially distributed data"* (p2L29–30). And what is the most (or at least more) efficient approach? Isn't the dS2 model essentially doing the same or what exactly is the difference?

- *"Many aspects of distributed modelling [...] require a high number of runs which further increases the computational demand. An efficient distributed conceptual model to tackle these kind of issues is currently lacking."* (p2L34–35). With computational infrastructure nowadays (HPCs, cloud computing) it is already possible to conduct hundred thousands or even millions of iterations of conceptual models within acceptable time frames, depending on resolution and spatio-temporal domain (see examples in Beven and Binley, 2014). Why then do we need more computationally efficient *conceptual* models? To conduct billions of runs in the same time? I doubt that this would substantially increase the value of uncertainty analysis and parameter estimation.

- *"Low-level languages generally perform faster calculations, but this comes at the price of user-friendliness and ease-of-use."* (p9L3–4). I don't understand this relationship.

Doesn't user-friendliness and ease-of-use depend on the interface between model and user, i.e. the structure of input and output files, how to run the model etc.? You can also program complicated models with high-level languages. And what about users, who prefer GUIs? For them a purely command line-based model will never be considered user-friendly. Moreover, user-friendly models, in my opinion, always contain a good explanation of file structures and a simple test case, which is unfortunately not the case for dS2.

- *"Simulating Europe for three months at hourly time steps and at a resolution of 5x5 km2"* (p9L18). Does it make sense to do that with dS2? Otherwise I don't understand why you choose such an unrealistic comparison.

- Section 3.2 (Adaptive time stepping): I think this sub-section can be hard to follow for anyone who is not familiar with numerical integration. E.g. what is the *Runge-Kutta scheme*, what are *fourth and fifth order estimations*? Maybe you can add a paragraph with a (very) short introduction to numerical integration, how it basically works, and why it is needed. It might also be worthwhile to stress the relevance in hydrological modelling as the issue is often neglected (most hydrological models just use explicit Euler with operator splitting). Some interesting papers about the topic: Clark and Kavetski (2010), Kavetski and Clark (2010), Kavetski and Clark (2011) and Schoups et al. (2010).

- Comment on numerical integration: As the dS2 is so extremely computationally efficient I wonder why you don't try the much more accurate implicit solvers. Of course they will increase computation time, but on the other hand will deliver much more accurate results (and potentially more reliable parameter and uncertainty estimations, see e.g. Kavetski and Clark, 2011). Maybe the GNU Scientific Library (`https://www.gnu.org/software/gsl/`) is worth a try (never tried by myself, but the website says *the interface was designed to be simple to link into very high-level languages, such as GNU Guile or Python*).

- *"[...] base the volume estimation on the mean discharge of the resulting shorter steps"* (p13L2). Is that indicated by $Q_{internal}$ in Fig. 7? Please clarify that in the figure, as $Q_{internal}$ itself is not explicitly defined.

- *"the current version of the model only outputs discharge at the end of the time step"* (p13L8–9). But internally $Q_{internal}$ is used, i.e. $\int_{t-1}^{t} Q_{internal}$ to calculate $S_t$?

- Fig. 8: In the text you write *response of the model to each parameter*, but the model contains more than the five parameters shown in Fig. 8?! Please be more explicit about what you mean.

- Section 4 (Parameter sensitivity → Sobol' sensitivity analysis): The Sobol' method requires that parameters are independent but in relation to Fig. 8 it is mentioned that some parameters are correlated. This will distort the results of the sensitivity analysis. Or produce strange results as can be seen in Fig. 8, namely that in some cases the total

effect is smaller than the main effect (or does the size of the bars reflect main + total effect? Please clarify). It seems also strange that for $KGE - \beta$ the total effect is always zero (if my interpretation of the bars is right). In any case, a possible workaround for correlated parameters is presented by Kucherenko, Tarantola and Annoni (2012) (not really an ad-hoc implementation). However, to see if the effort is really necessary, please first check the correlations among parameters (e.g. via covariance matrix). You might also consider a different method for sensitivity analysis, which is not affected by correlations (see review papers for guidelines, e.g. Pianosi et al., 2016).

- *"These graphs also show that there are some parameter correlations influencing the results"* (p16L1). What exactly do you mean? The discrepancy between main and total effects I mentioned earlier?

- *"We see that, although the magnitude of the peak is not fully captured in the Rietholzbach and Andelfingen basins, the timing of the peak is well simulated in all three basins."* (p17L8–9). But for Andelfingen the simulated peak occurs several hours before the measured peak, even with the routing module (the difference between routing and no routing module seems to be less than the difference between observations and routing module). As I see, this specific issue has also been addressed by the other reviewer.

**3 Technical corrections**

| | |
|---|---|
| p1L1 | efficiency $\rightarrow$ efficient |
| p1L8–9 | *"at high temporal and spatial resolution"*: one of the two occurrences of that phrase can be scratched. |
| p3L23 | *"in a flexible and efficient manner"*: also mentioned twice in the same sentence. |
| p10L11 | *"Since this concept [...]"*: As here a new sub-section starts, please specify explicitly what is meant by *this concept*. I.e. the dS2 model concept or the concept of adaptive time stepping (as this is the heading of the sub-section)? |
| Fig. 6 | I think this figure is rather hard to understand (also because of the two meanings of the Y axis). In panel 1) what do the + signs represent? |
| Figures | All panels must be indicated with brackets around lower case letters, e.g. (a), (b). See house standards (the proof reading will care about that anyway). |

| | |
|---|---|
| p14L1 | *"It is striking that the three g(Q) parameters [...]"*: a short reminder what the three parameters are (of the four shown in Fig. 8) would be helpful. |
| p14L8 | *"due to the large equifinality regions"*: it would be good to help readers with the interpretation, e.g. by adding to the mentioned sentence something like *"shown in dark blue"*. |
| p14L9 | full stop. |
| Text in general | Note the difference between different dashes, i.e. '-' and '–' (and sometimes '—'). I think throughout the text always '-' is used, where it is sometimes not appropriate, e.g. it should read *Kling–Gupta efficiency* (instead of *Kling-Gupta*), see GMD guidelines. |
| Fig. 10 | Please describe in the caption, what the KGE value in brackets stands for and what are the *"selected best runs"* (figure should be self-explaining). |

**References**

Beven, Keith and Andrew Binley (2014). 'GLUE: 20 years on'. In: *Hydrological Processes* 28.24, pp. 5897–5918. DOI: 10.1002/hyp.10082.

Clark, Martyn P. and Dmitri Kavetski (2010). 'Ancient numerical daemons of conceptual hydrological modeling: 1. Fidelity and efficiency of time stepping schemes'. In: *Water Resources Research* 46.10, W10510. DOI: 10.1029/2009WR008894.

Fenicia, Fabrizio et al. (2016). 'From spatially variable streamflow to distributed hydrological models: Analysis of key modeling decisions'. In: *Water Resources Research* 52.2, pp. 954–989. DOI: 10.1002/2015WR017398.

Kavetski, Dmitri and Martyn P. Clark (2010). 'Ancient numerical daemons of conceptual hydrological modeling: 2. Impact of time stepping schemes on model analysis and prediction'. In: *Water Resources Research* 46.10, W10511. DOI: 10.1029/2009WR008896.

— (2011). 'Numerical troubles in conceptual hydrology: Approximations, absurdities and impact on hypothesis testing'. In: *Hydrological Processes* 25.4, pp. 661–670. DOI: 10.1002/hyp.7899.

Kucherenko, S., S. Tarantola and P. Annoni (2012). 'Estimation of global sensitivity indices for models with dependent variables'. In: *Computer Physics Communications* 183.4, pp. 937–946. DOI: 10.1016/j.cpc.2011.12.020.

Metcalfe, Peter, Keith Beven and Jim Freer (2015). 'Dynamic TOPMODEL: A new implementation in R and its sensitivity to time and space steps'. In: *Environmental Modelling & Software* 72, pp. 155–172. DOI: 10.1016/j.envsoft.2015.06.010.

Pianosi, Francesca et al. (2016). 'Sensitivity analysis of environmental models: A systematic review with practical workflow'. In: *Environmental Modelling & Software* 79, pp. 214–232. DOI: 10.1016/j.envsoft.2016.02.008.

Schoups, G. et al. (2010). 'Corruption of accuracy and efficiency of Markov chain Monte Carlo simulation by inaccurate numerical implementation of conceptual hydrologic models'. In: *Water Resources Research* 46.10, W10530. DOI: 10.1029/2009WR008648.

---

## Author Comment (AC2) · 19 Dec 2019

We thank Anonymous Referee #2 (AR2) for providing a detailed and well underpinned review. Below, we will respond to the points made by AR2 in the same order as in the review: starting with the general comments, followed by the specific comments and finally the technical corrections.

**General comments**

Summary

We thank AR2 for the kind words on the manuscript and model, and hope to clarify the problems raised in the following sections.

Originality

Whether or not the hydrological community needs more models has been a long debated topic. We should stress that we would not have developed the models without the belief that it fills an important niche. There are a few important things that make this model stand out from the already existing conceptual models, and we believe that we can learn from the model.

First of all, the underlying concept of the dS2 model, the simple dynamical systems (SDS) approach, is based directly on discharge observations (Kirchner, 2009). The SDS approach is popular and sparked many scientific studies/discussions: the original paper is cited over 300 times according to Web of Science. The dS2 model allows us to apply a proven concept in a way that it supports larger catchments with more spatial variability. In contrast to this, the many bucket based conceptual models are based on our conceptual understanding of the hydrological system rather than observations. Therefore, dS2 relies on a different philosophy than the vast majority of conceptual models. Additionally, the SDS approach calculates changes in storage only expressed in terms of discharge rather than absolute storage values. As this is different from the large majority of conceptual models, dS2 can give new insights into the hydrological response.

Furthermore, as this is a computationally efficient hydrological model (though we do not state that this is the most efficient hydrological model, see our reply in the "Efficiency" section), it allows for relatively cheap sensitivity uncertainty and sensitivity studies. This is also something the majority of existing conceptual models cannot achieve.

Another point made by AR2 is that dS2 is solely a rainfall-runoff transformation tool. Simply stating, this is correct, but the same point can be made for every model – when discharge is the main variable of interest. Furthermore, the term rainfall-runoff transformation tool can be perceived as something 'black box', while there is definitely a physical hydrological understanding underlying the concept of this model. This ties directly to another point made by AR2: that this model will not help understanding processes within a catchment. This is correct, as these processes are all indirectly captured by the sensitivity function. However, the model can definitely help with understanding how the discharge of a catchment will respond under different scenarios, and how this relates to storage in a catchment. We understand how the current formulation of the introduction might suggest that dS2 will help with understanding specific hydrological processes, while we refer to the hydrological response. This will be clarified in the next version of the manuscript.

In order to more explicitly state the niche of this model, we propose to slightly alter the title of the manuscript to: A distributed simple dynamical systems approach (dS2 v1.0) for computationally efficient hydrological modelling at high spatial and temporal resolution.

Efficiency

AR2 correctly states that we sacrificed some computational efficiency to ensure code readability. Maybe the word "efficiency" in the title suggests that we tried to build the most computationally efficient conceptual hydrological model, but this was not our main goal. If this were the case, it would indeed make a lot more sense to use compiled languages such as C++ or Fortran. Of course, we do wanted to utilize the characteristics of the SDS approach in order to have a model that is computationally efficient. Within the Python programming language, we tried to write the code as efficient as possible. Therefore, the model heavily depends on the Numpy library, which utilizes C libraries for its calculations. This library allows for example the vectorization of functions, something that is not supported by the default Python functions.

Furthermore, as Python is continuously getting more and more popular, it allows other users to understand, improve and/or extend the model, or change part of the code depending on the research question at hand. We see models not as static entities, but rather as flexible environments in which ALL elements (not just parameterizations but also the numerical "core") can easily be adapted. A widely known language such as Python is much better tailored to this task . This could indeed also be done with using Python as an interface, but given the previous arguments we have chosen Python as the preferred programming language. We will include this reasoning as well the manuscript.

Applicability

We thank AR2 for their compliments on the readability of the code. Based on the comments and testing by the reviewer, we understand and agree that application of the model is currently rather ambiguous. It should be noted that we have several MSc students currently working with the model, and they don't experience any difficulties once the input data is provided. Therefore, we already started working on an example, with some documentation on how to setup and run the model. This can be found in the GitHub repository of the dS2 model (https://github.com/JoostBuitink/dS2/tree/master/example_application, see model_guide.ipynb). Please note that this is still work in progress, and will also be evaluated by the MSc students who are working with the model.

Sensitivity analysis

AR2 is correct in stating that we did not account for parameter interaction in the applied Sobol' sensitivity analysis, even though we are aware that correlation exists. This is also the reason why we both show the main and total effect in the figures showing the results from this analysis, as they give an insight into the individual and combined effects of the parameters. We will take a look at the covariance matrix (as stated later in the specific comments) and decide which method for sensitivity analysis is best for our case.

**Specific comments**

- p2L26-27: AR2 has a valid point, we also wanted to refer here to data available both at high spatial and temporal data such as radar data. We understand that this is currently not clearly stated, and we will add this to the next version of the manuscript.
- p2L29-30: The second sentence refers to the first part of the sentence before this one: applying a lumped model to each pixel, so not utilizing function vectorization for example. This is indeed not clear in the text, and we will rephrase this to improve the connection between the two parts.

- p2L34-35: AR2 is correct that HPCs and other solutions allow for many iterations within reasonable amounts of time. However, an efficient distributed conceptual model such as dS2 allows a user to do many runs on their own computer, without requiring HPCs and the costs related to the usage of these infrastructures. We agree that increasing the number of iterations from millions to billions is unlikely to add anything regarding uncertainty analyses and parameter estimations.
- p9L3-4: Here we aim towards the popularity of Python and its flexibility in changing and adapting the model code. We understand the confusion, and will clarify this in the next version of the manuscript. As stated before, we agree that adding a simple test case with definitely help others to use dS2.
- p9L18: AR2 is right that this doesn't really make sense to do this with dS2. We added this value as a reference, as this is likely to speak more to the reader than just the number of pixels within the region of interest.
- Section 3.2: We agree with the suggestion of AR2 and will add some more explanation about the numerical integration and the common approaches in currently existing models.
- Comment on numerical integration: AR2 is correct that the computationally efficiency character of the model allows for more advanced numerical integration methods. There is a tradeoff however, since a large part of the current computation time is already consumed by the numerical integrator. After extensive stress testing of our custom time stepping scheme and comparisons with known implicit solvers, we are convinced that our integrator is able to produce accurate results. Furthermore, it gives the user some control over the numerical precision and number of additional time steps to reach this precision, something not all already established solvers allow.
- p13L2: This is indeed the case, and we will make sure this is better described in the next version of the manuscript.
- p13L8-9: $Q_{internal}$ is currently not used in the model at all, as we added it here only to check the water balance. We understand the origin of the confusion and make sure this is clarified in the next version of the manuscript.
- Fig. 8: AR2 correctly spotted that there are more than 5 parameters in the model. However, most other parameters are related to either numerical stability or administrative functions within the model. We will explicitly state that we are investigating the five parameters influencing the hydrological response (α, β, γ, ε, τ).
- Section 4: See our reply in the section *Sensitivity analysis* above. Regarding the sizes between the total and main effect: the total effect is always bigger than the main effect, as both bars start at 0. We understand that the current visualization might look like stacked bars, while they are in fact not. We will slightly shift both bars to depict that they are separate bars. For the KGE – β (bias), the total effect is indeed close to zero for four out of five parameters, as only the ε (evaporation correction parameter) affects the bias.
- p16L1: We do indeed mean the discrepancy between main and total effects, and we will clarify this in the next version of the manuscript
- p17L8-9: AR2 is correct, this text was still belonging to an old (incorrect) version of this graph. We will update the text and explanation accordingly.

**Technical corrections**

We thank AR2 for their suggestions for technical corrections. We will implement those in the next version of the manuscript.

---

## Author Response (AR2)

**Technical corrections after the report of 29-04-2020**

- We clarified that actual evaporation is required by the concept in Section 2.1.

- We rewritten the first paragraph of Section 3.2 to avoid confusion.

- The dashed green line is indeed not visible due to overlay, we have added this to the caption.

**Point-by-point response to reviews**

Below, we will give a point-by-point response the reviews, with the comments from the reviews in black, and our reply in blue.

**Response to AR1**

Dear authors, thank you for submitting your paper. In the following, you will find my comments, suggestions, and opinion of your contribution.

**Summary**
    The paper "A distributed simple dynamical systems approach (dS2 v1.0) for computationally efficient hydrological modelling" is about a distributed hydrological modelling based on the simple dynamical system approach, SDS (Kirchner, 2009). The model is grid-based (cell of 1km2) and enables rainfall-runoff simulations for mesoscale basins (up to 2000 km2) at high temporal and spatial resolution with a new 3-parameter discharge sensitivity function. The model is first developed and assessed for synthetic experiment and then applied to the alpine Thur catchment (1700 km2). Authors used Monte-Carlo methodology to generate model parameter sets constrained within the synthetic parameter ranges. The model is complemented with snow and routing modules to take into account the lag-time.

**General comments**
    The manuscript is clearly written but has in its present form some major defaults. The authors put forward the novelty of the proposed approach. This is not right since the approach is not unique in its formulation. What is proposed is a simple dynamical system approach by Kirchner applied in its distributed form. Such model, however already exists (please check Adamovic et al. 2016, Journal of Hydrology). The latter was not cited in this article. Adamovic et al. (2016) implemented the SDS approach on Ardeche catchment (around 2400 km2) and developed the model called SIMPLEFLOOD that is computationally fast and coupled with a kinematic wave flow propagation module.
    We understand the concern from AR1 about the fact that we did not include a reference to SIMPLEFLOOD. In this new version of the manuscript, we have added a paragraph on SIMPLEFLOOD in the introduction. We have added reasoning on why we believe that dS2 is an important addition to the existing suite of hydrological models is still a valuable addition as a conceptual model (distributed using a grid, focus on computational efficiency, difference in numerical implementation, inclusion of additional processes such as snow for application to colder climates and/or more mountainous areas, and open source code).

**Specific comments**
    Here are, however, my short comments that I think can contribute to its better content once reformulating and updating the article.
    **Page 2. Line: 25** The authors mention continental-scale forcing datasets in relation with merged radar data, interpolated station data, and atmospheric reanalysis. It would be valuable for readers if they know in more details about actual spatial but also temporal scales of provided references.
    After the references, we have added a spatial and temporal scales on which these datasets are available.
    **Page 2. Line: 35** The phrase "An efficient distributed conceptual model to tackle these kinds of issues is currently lacking." is not correct. Adamovic et al. (JoH, 2016) already developed this approach at an hourly time step and applied it to the Ardeche catchment.
    See our reply above under "General comments".

**Page 3. Line: 25** Similar to the previous comment. The approach was already applied to the mesoscale catchment of 2400 km2 (Adamovic et al., 2016).

See our reply above under "General comments".

**Page 6. Line: 15** The introduction of a third parameter that avoids having a curvature that diverges computations is interesting even though it has its limitations since it seems that three parameters depend on each other.

It does indeed add some complexity to the two already correlated parameters $\alpha$ and $\beta$. As with any fitting of any curve, the fitting parameters will not be completely independent. This is not a specific disadvantage of the curve chosen. However, given that most previous studies showed a sensitivity function which could not be captured with a simple power–law, we believe that this is one of the most elegant solutions to cover known catchment behaviour.

**Page 8. Flow routing** Authors introduced simplified routing module that is not explicit but rather based on a simple temporal delay to the outlet. More sophisticated flow routing module is described in Adamovic et al. (2016) which takes into account the feedback effects.

See our reply above under "General comments". We present a conceptual model. Similar to the assumption of a single subsurface storage, we make a simplifying assumption that flow speed is constant across the stream network. We are fully aware of the fact that this is a crude assumption, but it is one that works well in terms of both model performance and computational efficiency. This concept is not new (width function, see Kirkby (1976)), and has been used in other studies as well (e.g. Franchini and O'Connell, 1996).

**Page 12. Closure of the water balance** In the model, the water balance seems closed only in conditions when AET=PET (humid conditions). If the model is to be applied on pan-European scales with different conditions, the question about the AET estimation should be addressed.

AR1 is correct that dS2 is currently not able to reduce evaporation during e.g. dry periods. This is why we rely on the user to provide accurate actual evaporation input data, as is stated in the Discussion. These datasets are now increasingly becoming available at temporal and spatial resolutions that dS2 runs at, so we think dS2 should be seen as a platform that can directly utilize this kind of data for runoff simulations. This approach (relying on input of AET rather than PET) is indeed fundamentally different from current modelling approaches.

**Page 13. Parameter sensitivity** It would be useful for a reader to know what the realistic parameter ranges are and how did you choose them since you applied the method on the synthetic catchment in this section.

The ranges presented in this figure show realistic parameter ranges, based on comparisons between the shape of the resulting discharge sensitivity function with previous studies using the simple dynamical systems approach. This information was indeed lacking, but is now added to the manuscript in the section about parameter sensitivity.

**Page 16. Example application** Line: 23. Did you also run the MC analysis with the fewer or higher number of iterations? Why did you choose 25000? Many authors used fewer iterations for higher-order parametric models (e.g. Tekleab et al. 2011).

We selected this number of runs to ensure good coverage of the parameter space. Initially the number was based on the study by Melsen et al. (2016a). As VIC is a higher-order parametric model, this number of runs should be sufficient for a model like dS2. We do not see a problem when using too many runs for a MC analysis, problems could indeed arise when one uses a too small sample size.

**Page 17. Line: 7** "Parameter set that performer best for the event". Is it the one that showed the highest KGE? If it is the case, what was the value?

This was indeed not correct in the previous version of the manuscript. We have updated the figure to ensure the correct runs are shown. The runs selected for the event are the runs with the best KGE performance over the total period showed in the other panels, where also the KGE values of these runs are shown.

**Page 17. Line: 9** "...the timing of the peak is well simulated in all three basins". It seems that for the Andelfingen catchment (Fig. 10e), there is a peak delay of around 6 h between the observation and simulation for this specific event. I would suggest reformulating this sentence and discussing what could be a reason for such peak delay since it is the largest explored catchment. Could it be that the simplified routing module shows the lack of performance in this case? Testing it in more catchments could give a more robust answer.

The text and the figures were not updated correctly in the previous version of the manuscript. In this new version of the manuscript, we have updated both the figure and text to better represent the output of the model. With the new run, the discrepancy in timing of the peak between observation and simulation is reduced. Since this basin contains many sub-basins,

we decided to keep this case study to only a single basin. We are currently applying the model to different basins to investigate the performance.

**Technical corrections**

**Figure 1.** Panels in capitals A, B, C instead of small letters. This will improve the text visibility.

To comply with the standards of GMD, we have decided to keep the panels in lower case letters.

**Figure 10 a.** I would suggest colouring the catchments for better visibility.

We have highlighted the shape of the three catchments shown in this figure.

**Figure 10 b, c and d.** I would suggest plotting these figures on a log-scale for better visibility and assessing the influence of evapotranspiration.

We tried plotting these panels on a log-scale, but this gave a distorted look on the model output, as much less emphasis is placed on the peaks. Since the aim of this figure is to visualize the model output, we decided to keep this figure with a normal axis.

**Figure 10.e** I would suggest not using the light colours for observations since it is not well readable (i.e. Mogelsberg catchment). Asterix or other dash types could be more appropriate in this case.

We have updated the figure and changed the linestyle of the observations to improve visibility.

**Response to AR2**

Note: some of our answers are largely copied from our reply to AR2 in the discussion.

**General comments**

**Summary** The manuscript A distributed simple dynamical systems approach (dS2 v1.0) for computationally efficient hydrological modelling by Buitink and colleagues introduces a distributed hydrological conceptual model oriented at the simple dynamical systems approach by Kirchner (2009). The model has been designed to simulate rainfall–runoff dynamics with high spatial and temporal resolution at small (100 km2) up to mesoscale (103 km2) catchments. The model should be straightforward to apply, computationally efficient, and the associated Python code is openly available and easy to understand and modify. In their manuscript the authors present a sensitivity analysis of model parameters and apply the model in the alpine Thur catchment. Most parts of the paper are well structured and good to understand. However, I see some serious problems with the paper (and the model) that should be resolved before final publication. Even more than for my concerns regarding originality and efficiency this holds true for the applicability of the model, as I will explain in the following paragraphs.

**Originality** As even the authors admit, many different conceptual hydrological models already exist. Therefore the question arises if we really need yet another model. My personal answer would be: maybe, if we can learn something from it. However, dS2 seems to me rather like a typical black box model and I don't see a way to learn anything about processes or the catchment. Even though the authors see their model as a valuable tool for educational purposes (page 20, line 17 ff.), other conceptual models based on multiple storages (or buckets) are, in my opinion, much more suitable for education. For instance, storages can be related to natural phenomena (e.g. surface, quick subsurface, and baseflow) and students can learn something about processes and parameter sensitivities (for an illustration, see e.g. Fenicia et al., 2016). As far as I can tell, dS2 serves solely as a very simple rainfall–runoff transformation tool and is therefore nothing new. I wonder how we can learn something about processes in a catchment, as the authors claim. To me this is not evident from the example application.

Whether or not the hydrological community needs more models has been a long debated topic. We should stress that we would not have developed the models without the belief that it fills an important niche. There are a few important things that make this model stand out from the already existing conceptual models, and we believe that we can learn from the model.

First of all, the underlying concept of the dS2 model, the simple dynamical systems (SDS) approach, is based directly on discharge observations (Kirchner, 2009), as is stated in the introduction. The SDS approach is popular and sparked many scientific studies/discussions: the original paper is cited over 300 times according to Web of Science. The dS2 model allows us to apply a proven concept in a way that it supports larger catchments with more spatial variability. In contrast to this, the many bucket based conceptual models are based on our conceptual understanding of the hydrological system rather than observations. Therefore, dS2 relies on a different philosophy than the vast majority of conceptual models. Additionally, the

SDS approach calculates changes in storage only expressed in terms of discharge rather than absolute storage values. As this is different from the large majority of conceptual models, dS2 can give new insights into the hydrological response.

Furthermore, as this is a computationally efficient hydrological model (though we do not state that this is the most efficient hydrological model, see our reply below in the "Efficiency" section), it allows for relatively cheap sensitivity uncertainty and sensitivity studies. This is also something the majority of existing conceptual models cannot achieve.

Another point made by AR2 is that dS2 is solely a rainfall-runoff transformation tool. Simply stating, this is correct, but the same point can be made for every model — when discharge is the main variable of interest. Furthermore, the term rainfall–runoff transformation tool can be perceived as something 'black box', while there is definitely a physical hydrological understanding underlying the concept of this model. This ties directly to another point made by AR2: that this model will not help understanding processes within a catchment. This is correct, as these processes are all indirectly captured by the sensitivity function. However, the model can definitely help with understanding how the discharge of a catchment will respond under different scenarios, and how this relates to storage in a catchment. It can also help to investigate whether runoff dynamics in a particular catchment are consistent with those from a simple dynamical system. We have made some changes in the introduction to better state the aim of this model.

In order to more explicitly state the niche of this model, we have also slightly altered the title of the manuscript to: A distributed simple dynamical systems approach (dS2 v1.0) for computationally efficient hydrological modelling at high spatio-temporal resolution.

**Efficiency** The authors stress at many occasions the computational efficiency of their model. On the other hand, computational efficiency has been sacrificed in favour of code readability by implementing it in Python, a widely used scripting language that is known for its well structured syntax in contrast to much more efficient compiled languages such as C++ or Fortran. Therefore, in my view the argument of computational efficiency, which is even advertised in the title of the manuscript, does not hold. A good compromise would have been to outsource the most expensive parts of code to a compiled language and use Python as an interface, as has been done for other models (e.g. TOPMODEL, for which an R interface exists, while the core model is written in Fortran, see Metcalfe, Beven and Freer, 2015)

AR2 correctly stated that we sacrificed some computational efficiency to ensure code readability. Maybe the word "efficiency" in the title suggests that we tried to build the most computationally efficient conceptual hydrological model, but this was not our main goal. If this were the case, it would indeed make a lot more sense to use compiled languages such as C++ or Fortran. Of course, we do wanted to utilize the characteristics of the SDS approach in order to have a model that is computationally efficient. Within the flexible and widely-used Python programming language, we tried to write the code as efficient as possible. Therefore, the model heavily depends on the Numpy library, which utilizes C libraries for its calculations. This library allows for example the vectorization of functions, something that is not supported by the default Python functions.

Furthermore, as Python is continuously getting more and more popular, it allows other users to understand, improve and/or extend the model, or change part of the code depending on the research question at hand. We see models not as static entities, but rather as flexible environments in which ALL elements (not just parametrizations but also the numerical "core") can easily be adapted. A widely known language such as Python is much better tailored to this task . This could indeed also be done with using Python as an interface, but given the previous arguments we have chosen Python as the preferred programming language. We have added this reasoning to Section 3.1 to better explain our reasoning behind choosing Python as the programming language.

**Applicability** I acknowledge that the code of the model is indeed well readable, even for persons with little Python experience (including me). I only wonder how I should apply the model. The authors present an example application in their manuscript but no example data to test the model. What is even more, the input and output file structures are not well explained, neither in their submitted manuscript nor in the assets or the github repository. In that way I don't know how to initialise and apply the model. This is probably the most severe flaw of the presented work. In my opinion it is also violating the core principles of GMD's model code and data policy.

We have created a brief model manual explaining the required model input and data formats. This manual includes pieces of code to show how one can setup, run, and analyse output of dS2. The manual can be found in the GitHub repository of the dS2 model (https://github.com/JoostBuitink/dS2/tree/master/example_application) and will extended and be kept up to date with possible future changes to the model.

**Sensitivity analysis** I see a problem with the Sobol' method for variance-based sensitivity analysis that the authors applied. This method may not be applied in case parameters are correlated (which is the case, as is stated in the text). I will provide some more reasoning and suggestions in the specific comments.

We are aware that the Sobol' method works optimally in the case where parameter are not correlated. This is not exactly the case in our model, yet we still decide to use this method since it is the most widely used method to investigate the parameter sensitivity. Furthermore, the aim of this sensitivity analysis is to give a first impression of the influence of the parameters on performance metrics. To give an idea of the effect of parameter interaction, we also show the "total effect", but will focus on the "main effect" for this study.

**Specific comments**

In the following I will provide some specific comments. Quotations from the paper appear as emphasised text. Specific passages in the text are referenced as pmLn, meaning page m, line n.

– "There is a growing need for easy-to-apply models that can utilize the potential of spatially distributed input data" (p2L26–27). Many (most?) models already utilize spatially distributed information. I also think the demand for easy-to-apply models always has been present (whereas the judgement of what is easy-to-apply is rather subjective).

We have added a spatial and temporal resolution to this paragraph to clearly state that we aim at data at both high spatial and temporal resolutions. With the addition of the manual, the model should be easy–to–apply.

– "water is most often transferred to the outlet as a post-processing function. This is, however, not necessarily the most computationally efficient way to deal with spatially distributed data" (p2L29–30). And what is the most (or at least more) efficient approach? Isn't the dS2 model essentially doing the same or what exactly is the difference?

The sentence was referring to the first part of the sentence before that. We have rephrased this part to clearly state what we mean with the "most computationally efficient way."

– "Many aspects of distributed modelling [...] require a high number of runs which further increases the computational demand. An efficient distributed conceptual model to tackle these kind of issues is currently lacking." (p2L34–35). With computational infrastructure nowadays (HPCs, cloud computing) it is already possible to conduct hundred thousands or even millions of iterations of conceptual models within acceptable time frames, depending on resolution and spatio-temporal domain (see examples in Beven and Binley, 2014). Why then do we need more computationally efficient conceptual models? To conduct billions of runs in the same time? I doubt that this would substantially increase the value of uncertainty analysis and parameter estimation.

This is correct, yet a model like dS2 allows users to do these computationally demanding on their own machines, rather than using HPCs (and the costs related to that). We have rephrased this part to clarify our point.

– "Low-level languages generally perform faster calculations, but this comes at the price of user-friendliness and ease-of-use." (p9L3–4). I don't understand this relationship. Doesn't user-friendliness and ease-of-use depend on the interface between model and user, i.e. the structure of input and output files, how to run the model etc.? You can also program complicated models with high-level languages. And what about users, who prefer GUIs? For them a purely command line-based model will never be considered user-friendly. Moreover, user-friendly models, in my opinion, always contain a good explanation of file structures and a simple test case, which is unfortunately not the case for dS2.

We have rewritten this part to clearly argue why we believe that Python is a solid programming language for this model. Together with the new model manual, it is easier for users to apply this model in their own research areas.

– "Simulating Europe for three months at hourly time steps and at a resolution of 5x5 km2" (p9L18). Does it make sense to do that with dS2? Otherwise I don't understand why you choose such an unrealistic comparison.

We added this value as an reference, as this is likely to speak more to the reader than just the number of pixels. We have slightly rephrased the sentence to make this clear.

– Section 3.2 (Adaptive time stepping): I think this sub-section can be hard to follow for anyone who is not familiar with numerical integration. E.g. what is the Runge-Kutta scheme, what are fourth and fifth order estimations? Maybe you can add a paragraph with a (very) short introduction to numerical integration, how it basically works, and why it is needed. It might also be worthwhile to stress the relevance in hydrological modelling as the issue is often neglected (most hydrological models just use explicit Euler with operator splitting). Some interesting papers about the topic: Clark and Kavetski (2010), Kavetski and Clark (2010), Kavetski and Clark (2011) and Schoups et al. (2010).

We have added a brief introduction on numerical integration in Section 3.2, and how our approach compares to common approaches in hydrological modelling.

– Comment on numerical integration: As the dS2 is so extremely computationally efficient I wonder why you don't try the much more accurate implicit solvers. Of course they will increase computation time, but on the other hand will deliver much more accurate results (and potentially more reliable parameter and uncertainty estimations, see e.g. Kavetski and Clark, 2011). Maybe the GNU Scientific Library (https://www.gnu.org/software/gsl/) is worth a try (never tried by myself, but the website says the interface was designed to be simple to link into very high-level languages, such as GNU Guile or Python).

This could be a possibility, yet we believe that after extensive (stress) testing our custom numerical solver is able to accurately and efficiently solve the differential equation. One could opt for more complex and therefore more computationally demanding solvers, but question arises whether this is worth the additional computational demand.

– "[...] base the volume estimation on the mean discharge of the resulting shorter steps" (p13L2). Is that indicated by Qinternal in Fig. 7? Please clarify that in the figure, as Qinternal itself is not explicitly defined.

We have clarified the meaning of Qinternal, both in the text and in the caption.

– "the current version of the model only outputs discharge at the end of the time step" (p13L8–9). But internally Qinternal is used, i.e. $\int_{t-1}^{t} Q_{internal}$ to calculate St?

This is not correct, and we have clarified this in the text in Section 3.3.

– Fig. 8: In the text you write response of the model to each parameter, but the model contains more than the five parameters shown in Fig. 8?! Please be more explicit about what you mean.

We have explicitly stated which parameters we are investigating, as the majority of parameters mostly influence input/output operations and numerical stability.

– Section 4 (Parameter sensitivity, Sobol' sensitivity analysis): The Sobol' method requires that parameters are independent but in relation to Fig. 8 it is mentioned that some parameters are correlated. This will distort the results of the sensitivity analysis. Or produce strange results as can be seen in Fig. 8, namely that in some cases the total effect is smaller than the main effect (or does the size of the bars reflect main + total effect? Please clarify). It seems also strange that for KGE – $\beta$ the total effect is always zero (if my interpretation of the bars is right). In any case, a possible workaround for correlated parameters is presented by Kucherenko, Tarantola and Annoni (2012) (not really an ad-hoc implementation). However, to see if the effort is really necessary, please first check the correlations among parameters (e.g. via covariance matrix). You might also consider a different method for sensitivity analysis, which is not affected by correlations (see review papers for guidelines, e.g. Pianosi et al., 2016).

As mentioned above, this analysis is performed to give an impression of the parameter sensitivity, hence why we also focus on the main effect, and not on the total effect. We have also added the limitations of this method with respect to the correlated parameters of dS2.

– "These graphs also show that there are some parameter correlations influencing the results" (p16L1). What exactly do you mean? The discrepancy between main and total effects I mentioned earlier?

We did indeed mean this discrepancy, and we have clarified this in the text.

- "We see that, although the magnitude of the peak is not fully captured in the Rietholzbach and Andelfingen basins, the timing of the peak is well simulated in all three basins." (p17L8–9). But for Andelfingen the simulated peak occurs several hours before the measured peak, even with the routing module (the difference between routing and no routing module seems to be less than the difference between observations and routing module). As I see, this specific issue has also been addressed by the other reviewer

  We have updated both the text and the figure as they were old (incorrect) version. This new figure and explanation in the text fixes and addresses the issues raised.

**Technical corrections**

  We have implemented all technical corrections made by AR2.

**List of relevant changes**

- The introduction is slightly altered so the goal of dS2 is more clearly stated. The introduction also highlights the SIMPLEFLOOD model to and states how dS2 is different from SIMPLEFLOOD.

- The reasoning and implications of the chosen programming language is now more clearly formulated.

- We have added a paragraph which shortly explains the importance of numerical solvers, including a brief description of higher order numerical solvers.

- A model manual was previously missing: we have created a brief model guide to explain what input data is required to setup the model, and how to run the model and analyse the output. This manual will be kept up-to-date on the GitHub page of the model (https://github.com/JoostBuitink/dS2/tree/master/example_application).

- We have slightly altered the title to better state the gap dS2 tries to fill.

- We have improved Figure 9, to better depict the difference between the main and total effects.

- We have improved Figure 10 by: highlighting the basins of interest, correcting the model runs shown in panel (e), adjusting the linestyle in panel (e) to make the observations better visible, and improved the caption of the figure.

[revised manuscript text omitted]